# Interpreting Neurons in Deep Vision Networks with Language Models

**Nicholas Bai**[*]                                                                        *nicholaszybai@gmail.com*
*UC San Diego*

**Rahul A. Iyer**[*]                                                                             *rai396@utexas.edu*
*UT Austin*

**Tuomas Oikarinen**                                                                     *toikarinen@ucsd.edu*
*UC San Diego*

**Akshay Kulkarni**                                                                       *a2kulkarni@ucsd.edu*
*UC San Diego*

**Tsui-Wei Weng**                                                                             *lweng@ucsd.edu*
*UC San Diego*

**Reviewed on OpenReview:** *https://openreview.net/forum?id=x1dXvvElVd*

## Abstract

In this paper, we propose Describe-and-Dissect (**DnD**), a novel method to describe the roles of hidden neurons in vision networks. **DnD** utilizes recent advancements in multimodal deep learning to produce complex natural language descriptions, without the need for labeled training data or a predefined set of concepts to choose from. Additionally, **DnD** is *training-free*, meaning we don't train any new models and can easily leverage more capable general purpose models in the future. We have conducted extensive qualitative and quantitative analysis to show that **DnD** outperforms prior work by providing higher quality neuron descriptions. Specifically, our method on average provides the highest quality labels and is more than 2× as likely to be selected as the best explanation for a neuron than the best baseline. Finally, we present a use case providing critical insights into land cover prediction models for sustainability applications. Our code and data are available at `https://github.com/Trustworthy-ML-Lab/Describe-and-Dissect`.

## 1 Introduction

Recent advancements in Deep Neural Networks (DNNs) within machine learning have enabled unparalleled development in multimodal artificial intelligence. While these models have revolutionized domains across image recognition and natural language processing, they haven't seen much use in various safety-critical applications, such as healthcare or ethical decision-making. This is in part due to their cryptic "black box" nature, where the internal workings of complex neural networks have remained beyond human comprehension. This makes it hard to place appropriate trust in the models and additional insight in their workings is needed to reach wider adoption.

Previous methods have gained a deeper understanding of DNNs by examining the functionality (also known as *concepts*) of individual neurons[1]. This includes works based on manual inspection (Erhan et al., 2009; Zhou et al., 2015; Olah et al., 2020; Goh et al., 2021), which can provide high quality description at the cost

---

[*]N. Bai and R. Iyer contributed equally to this work. Major work done in Summer outreach program at UC San Diego.
[1]We conform to prior works' notation and use "neuron" to describe a channel in CNNs.

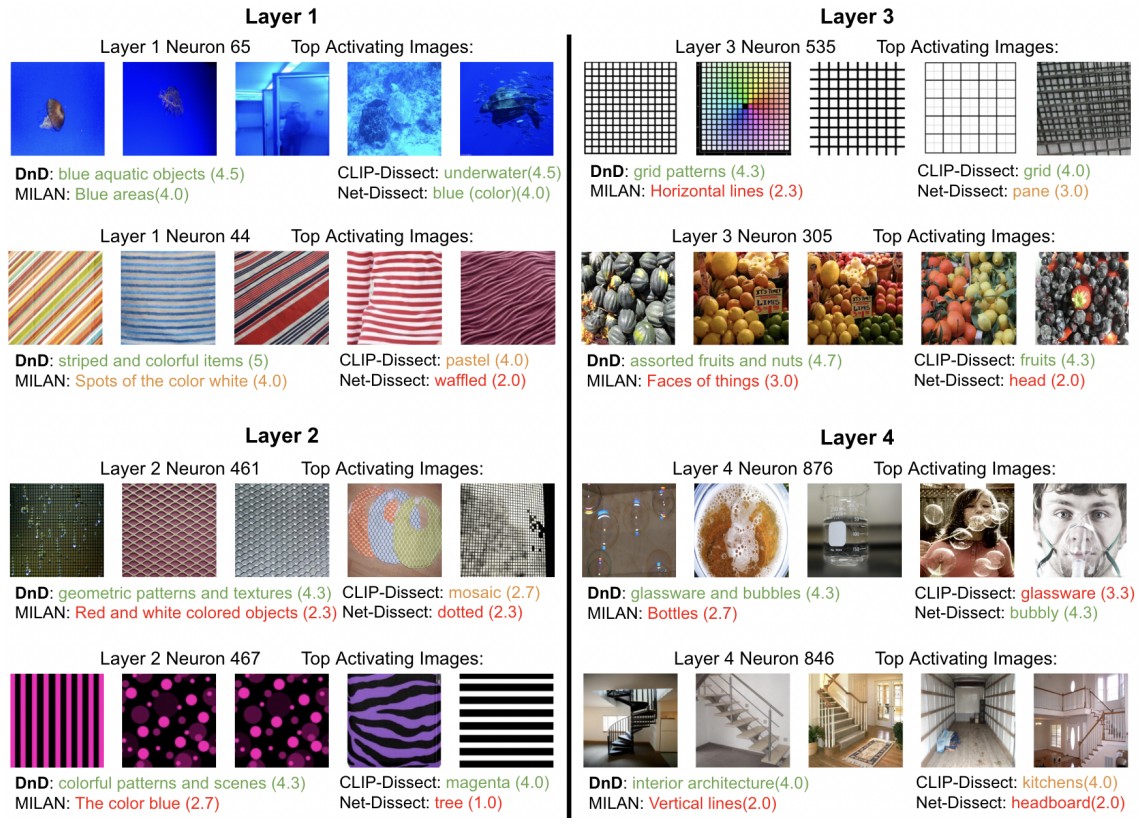

Figure 1: Neuron descriptions provided by our method (**DnD**) and baselines CLIP-Dissect (Oikarinen & Weng, 2023), MILAN (Hernandez et al., 2022)), and Network Dissection (Bau et al., 2017) for random neurons from ResNet-50 trained on ImageNet. We have added the average quality rating from our Amazon Mechanical Turk experiment described in Section 4.3 next to each label and color-coded the neuron descriptions by whether we believed they were accurate, somewhat correct or vague/imprecise.

of being very labor intensive. Alternatively, Network Dissection (Bau et al., 2017) automated this labeling process by creating the pixelwise labeled dataset, *Broden*, where fixed concept set labels serve as ground truth binary masks for corresponding image pixels. The dataset was then used to match neurons to a label from the concept set based on how similar their activation patterns and the concept maps were. While earlier works, such as Network Dissection, were restricted to an annotated dataset and a predetermined concept set, CLIP-Dissect (Oikarinen & Weng, 2023) offered a solution by no longer requiring labeled concept data, but still requires a predetermined concept set as input. By utilizing OpenAI's CLIP model, CLIP-Dissect matches neurons to concepts based on their activations in response to images, allowing for a more flexible probing dataset and concept set compared to previous works.

However, these methods still share a major limitation: Concepts detected by certain neurons, especially in intermediate layers, prove to be difficult to encapsulate using the simple, often single-word descriptions provided in a fixed concept set. MILAN (Hernandez et al., 2022) sought to enhance the quality of these neuron labels by providing generative descriptions, but their method requires training a new description model from scratch to match human explanations on a dataset of neurons. This leads to their proposed method being more brittle and often performs poorly outside its training data.

To overcome these limitations, we propose Describe-and-Dissect (abbreviated as **DnD**) in Section 3, a pipeline to *dissect* DNNs by utilizing an image-to-text model to *describe* highly activating images for corresponding neurons. The descriptions are then semantically combined by a large language model, and finally refined with synthetic images to generate the final concept of a neuron. We conduct extensive qualitative and quantitative analysis in Section 4 and show that Describe-and-Dissect outperforms prior work by providing high quality

Table 1: Comparison of existing automated neuron labeling methods and our method Describe-and-Dissect (**DnD**). Green and boldfaced **Yes** or **No** indicates the **desired property** for a column. **DnD** has all the **desired properties** while existing work has some limitations.

| Method \ property | Requires Concept Annotations | Training Free | Generative Natural Language Descriptions | Uses Spatial Activation Information | Can easily leverage better future models |
|---|---|---|---|---|---|
| Network Dissection (Bau et al., 2017) | Yes | **Yes** | No | **Yes** | No |
| MILAN (Hernandez et al., 2022) | Training only | No | **Yes** | **Yes** | No |
| CLIP-Dissect (Oikarinen & Weng, 2023) | **No** | **Yes** | No | No | **Yes** |
| FALCON (Kalibhat et al., 2023) | **No** | **Yes** | No | **Yes** | **Yes** |
| **DnD (This work)** | **No** | **Yes** | **Yes** | **Yes** | **Yes** |

neuron descriptions. Specifically, we show that Describe-and-Dissect provides more complex and higher-quality descriptions (up to 2-4× better) on intermediate layer neurons than other contemporary methods in a large scale user study. Example descriptions from our method are displayed in Figure 1. Additionally, we present a use-case study demonstrating **DnD**'s ability to interpret and improve upon current sustainability models in Section 5.

## 2 Background and related work

### 2.1 Neuron Interpretability Methods

Network Dissection (Bau et al., 2017) is the first method developed to automatically describe individual neurons' functionalities. The authors first defined the densely-annotated dataset *Broden*, denoted as $\mathcal{D}_{\text{Broden}}$, as a ground-truth concept mask. The dataset is composed of various images $x_i$, each labeled with concepts $c$ at the pixel-level. This forms a ground truth binary mask $L_c(x_i)$ which is used to calculate the intersection over union (IoU) score between $L_c(x_i)$ and the binary mask from the activations of the neuron $k$ over all images $x_i \in \mathcal{D}_{\text{Broden}}$, denoted $M_k(x_i)$: $\text{IoU}_{k,c} = \frac{\sum_{x_i \in \mathcal{D}_{\text{Broden}}} M_k(x_i) \cap L_c(x_i)}{\sum_{x_i \in \mathcal{D}_{\text{Broden}}} M_k(x_i) \cup L_c(x_i)}$. The concept $c$ is assigned to a neuron $k$ if $\text{IoU}_{k,c} > \eta$, where the threshold $\eta$ was set to 0.04. Intuitively, this method finds the labeled concept whose presence in the image is most closely correlated with the neuron having high activation. Extensions of Network Dissection were proposed by Bau et al. (2020) and Mu & Andreas (2020).

However, Network Dissection is limited by the need of concept annotation and the concept set is a closed set that may be hard to expand. To address these limitations, a recent work CLIP-Dissect (Oikarinen & Weng, 2023) utilizes OpenAI's multimodal CLIP (Radford et al., 2021) model to describe neurons automatically without requiring annotated concept data. They leverage CLIP to score how similar each image in the probing dataset $\mathcal{D}_{probe}$ is to the concepts in a user-specified concept set to generate a concept activation matrix. To describe a neuron, they compare the activation pattern of said neuron to activations of different concepts on the probing data, and find the concept that is the closest match using a similarity function, such as softWPMI. Another very recent work FALCON (Kalibhat et al., 2023) uses a method similar to CLIP-Dissect but augments it via counterfactual images by finding inputs similar to highly activating images with low activation for the target neuron, and utilizing spatial information of activations via cropping. However, they solely rely on cropping the most salient regions within a probing image to filter spurious concepts that are loosely related to the ground truth functionality labels of neurons. This approach largely restrict their method to local concepts while overlooking holistic concepts within images, as noted by Kalibhat et al. (2023). Their approach is also limited to single word/set of words descriptions that are unable to reach the complexity of natural language.

On the other hand, MILAN (Hernandez et al., 2022) is a different approach to describe neurons using natural language descriptions in a generative fashion. Note that despite the concept sets in CLIP-Dissect and FALCON being flexible and open, they cannot provide generative natural language descriptions like MILAN.

The central idea of MILAN is to train an image-to-text model from scratch to describe the neuron's role based on 15 most highly activating images. Specifically, it was trained on crowdsourced descriptions for 20,000 neurons from selected networks. MILAN can then generate natural language descriptions to new neurons by outputting descriptions that maximize the weighted pointwise mutual information (WPMI) between the description and the active image regions. One major limitation of MILAN is that the method require training a model to imitate human descriptions of image regions on relatively small training dataset, which may cause inconsistency and poor explanations further from training data. In contrast, our **DnD** is *training-free*, *generative*, and produces a *higher quality of neuron descriptions* as supported by our extensive experiments in Figure 1 and Table 3. A detailed comparison between our method and the baseline methods is shown in Table 1.

## 2.2   Leveraging Large Pretrained models

In our **DnD** pipeline, we are able to leverage recent advances in the large pre-trained models to provide high quality and generative neuron descriptions for DNNs in a *training-free* manner. Below we briefly introduce the Image-to-Text Model, Large Language Models and Text-to-Image Model used in our pipeline implementation. The first model is Bootstrapping Language-Image Pretraining (BLIP) (Li et al., 2022), which is an image-to-text model for vision-language tasks that generates synthetic captions and filters noisy ones, employing bootstrapping for the captions to utilize noisy web data. While our method can use any image-to-text model, we use BLIP in this paper for our Step 2 in the pipeline due to BLIP's high performance, speed, and relatively low computational cost. However, we note that our method can be easily adapted to leverage more advanced models in the future.

The second model is GPT-3.5 Turbo, which is a transformer model developed by OpenAI for understanding and generating natural language. It provides increased performance from other contemporary models due to its vast training dataset and immense network size. We utilize GPT-3.5 Turbo for natural language processing and semantic summarization in the Step 2 of our **DnD**. We use GPT-3.5 Turbo in this work as it's one of the SOTAs in LLMs and cheap to use, but our method is compatible with other future and more advanced LLMs. We provide a quantitative comparison analyzing the effect of GPT-3.5 Turbo, GPT-4.0, and LLaMA2 on **DnD**'s label quality in Appendix A.4.5 as well as evaluation on cost and usage limitations for each model.

The third model is Stable Diffusion (Rombach et al., 2022), which is a text-to-image latent diffusion model (LDM) trained on a subset from the LAION-5B database (Schuhmann et al., 2022). By performing the diffusion process over the low dimensional latent space, Stable Diffusion is significantly more computationally efficient than other diffusion models, such as DALLE (Ramesh et al., 2021). Due to its open availability, lower computational cost, and high performance, we employ Stable Diffusion for our image generation needs in the Step 3 of **DnD**.

## 3   Describe-and-Dissect: Methods

**Overview.**   In this section, we present Describe-and-Dissect (**DnD**), a comprehensive method to produce generative neuron descriptions in deep vision networks. Our method is training-free, model-agnostic, and can be easily adapted to utilize advancements in multimodal deep learning. **DnD** consists of three steps:

- **Step 1. Probing Set Augmentation:** Augment the probing dataset with attention cropping to include both global and local concepts. This helps better describe localized neuron activations.

- **Step 2. Candidate Concept Generation:** Generate initial concepts by describing highly activating images and subsequently summarize them into candidate concepts using GPT. This is the main step that generates a set of possible explanations.

- **Step 3. Best Concept Selection:** Generate new images based on candidate concepts and select the best concept based on neuron activations on these synthetic images with a scoring function. This refines the description by selecting the most accurate neuron description from descriptions produced in Step 2.

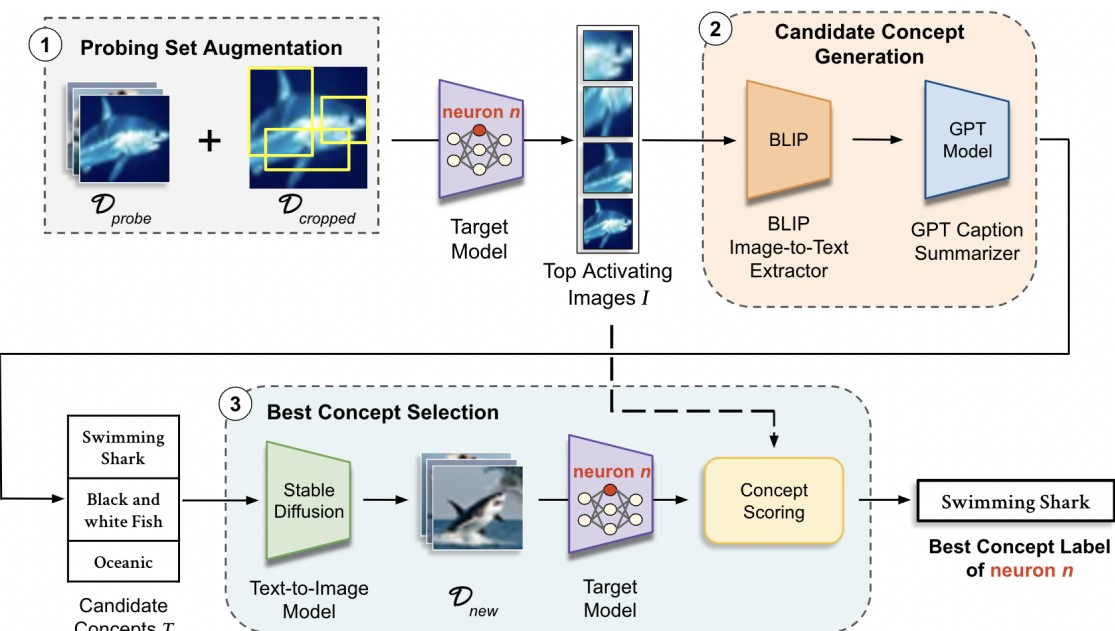

Figure 2: Overview of Describe-and-Dissect (**DnD**) algorithm. For a given target model, **DnD** consists of three important steps to identify neuron concepts (e.g. 'Swimming Shark' for neuron $n$).

An overview of Describe-and-Dissect (**DnD**) and these 3 steps are illustrated in Figure 2.

## 3.1 Step 1: Probing Set Augmentations

Probing dataset $\mathcal{D}_{probe}$ is the set of images we record neuron activations on before generating a description. As described in Section 2.1, one major limitation of Kalibhat et al. (2023) is the restriction to local concepts while overlooking holistic concepts within images, while one limitation of Oikarinen & Weng (2023) is not incorporating the spatial activation information. Motivated by these limitations, **DnD** resolves these problems by *augmenting* the original probing dataset with a set of attention crops of the highest activating images from the original probing dataset. The attention crops can capture the spatial information of the activations and we name this set as $\mathcal{D}_{cropped}$, shown in Figure 2. We discuss the implementation details of our attention cropping procedure in Appendix A.1.1 and perform an ablation study on its effects in Appendix A.4.1.

## 3.2 Step 2: Candidate Concept Generation

The top $K$ most highly activating images for a neuron $n$ are collected in set $I, |I| = K$, by selecting $K$ images $x_i \in \mathcal{D}_{probe} \cup \mathcal{D}_{cropped}$ with the largest $g(A_k(x_i))$. Here $g$ is a summary function (for the purposes of our experiments we define $g$ as the spatial mean) and $A_k(x_i)$ is the activation map of neuron $k$ on input $x_i$. We then generate a set of candidate concepts for the neuron with the following two part process:

- **Step 2A - Generate descriptions for highly activating images:** We utilize BLIP image-to-text model to generatively produce an image caption for each image in $I$. For an image $I_{j \in [K]}$, we feed $I_j$ into the base BLIP model to obtain an image caption.

- **Step 2B - Summarize similarities between image descriptions:** Next we utilize OpenAI's GPT-3.5 Turbo model to summarize similarities between the $K$ image captions for each neuron being checked. GPT is prompted to generate $N$ descriptions which identify and summarize the conceptual similarities between most of the BLIP-generated captions.

The output of **Step 2B** is a set of $N$ descriptions which we call "candidate concepts". We denote this set as $T = \{T_1, ..., T_N\}$. For the purposes of our experiments, we generate $N = 5$ candidate concepts unless otherwise mentioned. The exact prompt used for GPT summarization is shown in Appendix A.1.3.

### 3.3 Step 3: Best Concept Selection

The last crucial component of **DnD** is concept selection, which selects the concept from the set of candidate concepts $T$ that is most correlated to the activating images of a neuron. We first use the Stable Diffusion model (Rombach et al., 2022) from Hugging Face to generate images for each concept $T_{j \in [N]}$. Generating new images is important as it allows us to differentiate between neurons truly detecting a concept or just spurious correlations in the probing data. The resulting set of images is then fed through the target model again to record the activations of a target neuron on the new images. Finally, the candidate concepts are ranked using a concept scoring function, as discussed in Section 3.4.

**Concept Selection Algorithm**   The algorithm consists of 4 substeps. For each neuron $n$, we start by:

1. *Generate supplementary images.* Generate $Q$ synthetic images using a text-to-image model for each label $T_{j \in [N]}$. The set of images from each concept is denoted as $\mathcal{D}_j$, $|\mathcal{D}_j| = Q$. The total new dataset is then $\mathcal{D}_{new} = \bigcup_{j=1}^{N} \mathcal{D}_j = \{x_1^{new}, ..., x_{N \cdot Q}^{new}\}$, which represents the full set of generated images. For the purposes of the experiments in this paper, we set $Q = 10$.

2. *Feed new dataset $\mathcal{D}_{new}$, back into the target model and rank the images based on activation.* We then evaluate the activations of target neuron $n$ on images in $\mathcal{D}_{new}$ and compute the rank of each image in terms of target neuron activation. Given neuron activations $A_n(x_i^{new})$, we define $\mathcal{G}_n = \{g(A_n(x_1^{new})), ..., g(A_n(x_{N \cdot Q}^{new}))\}$ as the set of scalar neuron activations.

3. *Gather the ranks of images corresponding to concept $T_j$.* Let $\mathrm{Rank}(x; \mathcal{G})$ be a function that returns the rank of an element $x$ in set $\mathcal{G}$, such that $\mathrm{Rank}(x'; \mathcal{G}) = 1$ if $x'$ is the largest element in $\mathcal{G}$. For every concept $T_j$, we record the ranks of images generated from the concept in $\mathcal{H}_j$, where $\mathcal{H}_j = \{\mathrm{Rank}(g(A_n(x)); \mathcal{G}_n) \ \forall \ x \in \mathcal{D}_j\}$, and $\mathcal{H}_j$ is sorted in increasing order, so $\mathcal{H}_{j1}$ is the rank of the lowest ranking element.

4. *Assign scores to each concept.* The scoring function $score(\mathcal{H}_j)$ assigns a score to a concept using the rankings of the concept's generated images, and potential additional information. The concept with the best (highest) score in $T$ is selected as the concept label for the neuron. Concept scoring functions are discussed below in Section 3.4.

In simpler terms, the intuition behind this algorithm is that if a neuron $n$ encodes for a concept $c$, then the images generated to encapsulate that concept $c$ should cause the neuron $n$ to activate highly. While we only experiment with Best Concept selection within the **DnD** framework, it can be independently applied with other methods like Bau et al. (2017); Hernandez et al. (2022); Oikarinen & Weng (2023) to select the best concept out of their top-k best descriptions, which is another benefit of our proposed method. **DnD** can also be used without Best Concept Selection (Step 3) to reduce computational costs.

### 3.4 Scoring Function

For a given neuron, we use a scoring function to rate candidate concept accuracy during Best Concept Selection (Step 3). Simple metrics such as mean are heavily prone to outliers that result in skewed predictions so we propose a scoring function that weights the average rank of top activating images mapping to a candidate concept.

$$score(R_j, I, \mathcal{D}_j^t) = (N - \mathrm{Rank}(R_j)) \cdot E(I, \mathcal{D}_j^t)$$

Here, the average rank of images for candidate concept $j$, $\forall j \in \{1, ..., N\}$, is denoted $R_j$ and $\mathrm{Rank}(R_j)$ sorts the average rank $R_j$ of each candidate concept in increasing order. $E(I, \mathcal{D}_j^t)$ computes the average cosine similarity between image embeddings of $\mathcal{D}_j^t$ and $I$ using CLIP-ViT-B/16 (Radford et al., 2021), with $\mathcal{D}_j^t \subset \mathcal{D}_j$

for $t$ highest activating images (for our experiments, we use $t = 10$). In practice, $R_j$ is computed as the square of the ranks in top $\beta = 5$ ranking images for better differentiation between scores, $R_j = \{(R_j^i)^2; i \leq \beta\}$. Sections A.1.4 the details specifics behind the function. In Section A.1.5, we compare between various functions and show our algorithm works robustly with different options.

## 4 Experiments

In this section, we present extensive qualitative and quantitative analysis to show that **DnD** outperforms prior works by providing higher quality neuron descriptions. For fair comparison, we follow the setup in prior works to run our algorithm on the following two networks: ResNet-50 and ResNet-18 (He et al., 2016) trained on ImageNet (Russakovsky et al., 2015) and Place365 (Zhou et al., 2016) respectively. In Section 4.1, we qualitatively analyze **DnD** along with other methods on random neurons and show that our method provides good descriptions on these examples. Next in Section 4.2 we quantitatively show that **DnD** yields superior results to comparable methods. In Section 4.3, we show that our method outperforms existing neuron description methods in large scale crowdsourced studies. Finally in Section 4.4 we study the importance of critical steps in our pipeline by ablating away Generative Image Captioning (Step 2A) and Concept Selection (Step 3). Supplementary results are presented in the appendix, including method details in Section A.1, additional qualitative examples in Section A.2, comparison to MILANNOTATIONS in Section A.3, extensive ablation studies on each step of the **DnD** framework in Section A.4, an additional use case of **DnD** as an OOD classifier in Section A.6, and the capability to describe polysemantic neurons by producing multiple labels in Section A.7.

### 4.1 Qualitative evaluation

We qualitatively analyze results of randomly selected neurons from various layers of ResNet-50, ResNet-18, and ViT-B-16. Sample results are displayed in Figure 1 and Figures 9, 10, 11, 12, 13, 15, and 16 in the Appendix. We use the union of the ImageNet validation dataset and Broden as $\mathcal{D}_{probe}$ and compare to Network Dissection (Bau et al., 2017), MILAN (Hernandez et al., 2022), and CLIP-dissect (Oikarinen & Weng, 2023) as baselines. Labels for each method are color coded by whether we believe they are accurate, somewhat correct, or vague/imprecise. Compared to baseline models, we observe that **DnD** captures higher level concepts in a more semantically coherent manner. Specifically, methods such as CLIP-dissect and Network Dissection have limited expressability due to the use of restricted concept sets while MILAN produces labels confined to lower level concepts. Additionally, we find that **DnD** can express multiple concepts within a single label owing to its generative nature.

### 4.2 Quantitative Evaluation

**Final layer evaluation.** Here we follow Oikarinen & Weng (2023) to quantitatively analyze description quality on the last layer neurons, which have known ground truth labels (i.e. class name) to allow us to evaluate the quality of neuron descriptions automatically. To evaluate on accuracy, we measure the average CLIP and mpnet cosine similarity between ground-truth labels and generated descriptions on all output neurons of ResNet-50's final fully-connected layer. BERTScore metric measures the semantic interpretability of **DnD** descriptions. We focus this study on comparison with MILAN (Hernandez et al., 2022) as it is the other generative contemporary work in the baselines. Network Dissection (Bau et al., 2017) and CLIP-Dissect (Oikarinen & Weng, 2023) are not included in this comparison because these methods have concept sets where the "ground truth" class or other similar concepts can be included, giving them an unfair advantage to the methods without concept sets like MILAN and **DnD**. We reported the results for all of the neurons of ResNet-50's final fully-connected layer in Table 2. Our results show that **DnD** outperforms MILAN, producing labels that are significantly closer to the ground truths than MILAN's.

### 4.3 Crowdsourced experiment

**Setup.** Our experiment compares the quality of labels produced by **DnD** against 3 baselines: CLIP-Dissect, MILAN, and Network Dissection. For MILAN we used their most powerful *base* model in our experiments.

Table 2: **Textual similarity between predicted labels and ground truths on the fully-connected layer of ResNet-50 trained on ImageNet**. We can see **DnD** outperforms MILAN.

| Metric / Methods | MILAN | **DnD (Ours)** |
|:---:|:---:|:---:|
| CLIP cos | 0.7080 | **0.7598** |
| mpnet cos | 0.2788 | **0.4588** |
| BERTScore | 0.8206 | **0.8286** |

Table 3: **Averaged AMT results across four layers in ResNet-50 and ResNet-18**. We can see **DnD** outperforms existing methods on both ResNet-50 (ImageNet) and ResNet-18 (Places365). Our descriptions are consistently rated the highest and selected as the best more than twice as often as the best baseline.

| Metric / Method | ResNet-50 | | | | ResNet-18 | | | |
|---|---|---|---|---|---|---|---|---|
| | NetDissect | MILAN | CLIP-Dissect | **DnD (Ours)** | NetDissect | MILAN | CLIP-Dissect | **DnD (Ours)** |
| Mean Rating | 3.14±0.032 | 3.21±0.032 | 3.67±0.028 | **4.15±0.022** | 3.33±0.056 | 3.14±0.059 | 3.52±0.055 | **4.14±0.041** |
| Selected as Best | 12.71% | 13.29% | 23.11% | **50.89%** | 12.62 | 13.32% | 19.39% | **54.67%** |

We dissected both a ResNet-50 network pretrained on Imagenet-1K and ResNet-18 trained on Places365, using the union of ImageNet validation dataset and Broden (Bau et al., 2017) as our probing dataset. For both models we evaluated 4 of the intermediate layers (end of each residual block), with 200 randomly chosen neurons per layer for ResNet50 and 50 per layer for ResNet-18. Each neurons description was evaluated by 3 different workers. In total, 3000 human ratings were conducted, 2400 evaluations on ResNet-50 and 600 evaluations on ResNet-18.

The full task interface and additional experiment details are available in Appendix A.1.6. Workers were presented with the top 10 highest activating images of a neuron followed by four separate descriptions; each description corresponds to a label produced by one of the four methods compared. The descriptions are rated on a 1-5 scale, where a rating of 1 represents that the user "strongly disagrees" with the given description, and a rating of 5 represents that the user "strongly agrees" with the given description. Additionally, we ask workers to select the description that best represents the 10 highly activating images presented. For these highly activating images, we used the images calculated by our method. As our probing dataset is a superset of the image sets used by prior methods, we believe our model is the most accurate for determining images to visualize since the probing dataset encapsulates the most concepts.

**Results.** Table 3 shows the results of a large scale human evaluation study conducted on Amazon Mechanical Turk (AMT). Looking at "% time selected as best" as the comparison metric, our results show that **DnD** performs over 2× better than all baseline methods when dissecting ResNet-50 or ResNet-18, being selected the best of the four up to 54.67% of the time. In terms of mean rating, our method achieves an average label rating over 4.1 for both dissected models, whereas the average rating for the second best method, CLIP-Dissect, is only 3.67 on ResNet-50 and 3.52 on ResNet-18. Our method also significantly outperforms MILAN's *generative* labels, which averaged below 3.3 for both target models. Our method significantly outperforms existing methods in crowdsourced evaluation, and does this consistently across different models and layers.

### 4.3.1 MILANNOTATIONS evaluation

Though evaluation on hidden layers of deep vision networks can prove quite challenging as they lack "ground truth" labels, one resource to perform such task is the MILANNOTATIONS dataset (Hernandez et al., 2022), which collects annotated labels to serve as ground truth neuron explanations. We perform quantitative evaluation by calculating the textual similarity between a method's label and the corresponding MILANNOTATIONS. Our analysis in Section A.3 found that if every neuron is described with the same

constant concept "depictions", it will achieve better results than any explanation on the dataset, but this is not a useful nor meaningful description. We hypothesize this is due to high levels of noise and interannotator disagreement, leading to low textual similarity between descriptions and generic descriptions scoring highly. We conclude that this dataset is unreliable to serve as ground truths for comparing different methods.

## 4.4 Ablation Studies

### 4.4.1 DnD with fixed concept set

To analyze the importance of using a generative image-to-text model, we explore instead utilizing fixed concept sets with CLIP (Radford et al., 2021) to generate descriptions for each image instead of BLIP, while the rest of the pipeline remains unchanged (i.e. using GPT to summarize etc). For the experiment, we use CLIP-ViT-B/16, where we define $L(\cdot)$ and $E(\cdot)$ as text and image encoders respectively. From the initial concept set $\mathcal{S} = \{t_1, t_2, ...\}$, the best concept for image $I_m$ is defined as $t_l$, where $l = \text{argmax}_i(L(t_i) \cdot E(I_m)^{\top})$. Following CLIP-dissect (Oikarinen & Weng, 2023), we use $\mathcal{S} = 20\text{k}$ (20,000 most common English words)[2] and $\mathcal{D}_{probe} = \text{ImageNet} \cup \text{Broden}$.

To compare the performance, following Oikarinen & Weng (2023), we use our model to describe the final layer neurons of ResNet-50 (where we know their ground truth role) and compare description similarity to the class name that neuron is detecting, as discussed in Section 4.2. Results in Table 4 show that both methods perform similarly on the FC layer. In intermediate layers, we notice that single word concept captions from 20k significantly limit the expressiveness of **DnD**, suggesting generative image descriptions is important for our overall performance. Qualitative examples and notable failure cases of CLIP descriptions can be found under Appendix A.4.2.

Table 4: **Mean FC Layer Similarity of CLIP Captioning.** Utilizing a fixed concept set to caption activating images via CLIP (Radford et al., 2021), we compute the mean cosine similarity across fully connected layers of RN50. We find the performance of **DnD** w/ CLIP Captioning is slightly worse than BLIP generative caption.

| Metric / Methods | DnD (Ours) | DnD w/ CLIP Captioning | % Decline |
|---|---|---|---|
| CLIP cos | **0.7598** | 0.7583 | 0.197% |
| mpnet cos | **0.4588** | 0.4465 | 2.681% |
| BERTScore | **0.8286** | 0.8262 | 0.290% |

### 4.4.2 Effects of Concept Selection

We use 50 randomly chosen neurons from each of the 4 layers of ResNet-50 to conducted an ablation study on the impact of Best Concept Selection (Step 3) on the pipeline. Evaluations are conducted on the first candidate concept. Each neuron was evaluated twice yielding a total of 400 human ratings. Table 5 shows the effect of Best Concept Selection on the overall accuracy of **DnD**. We can see DnD performance is already high without Best Concept Selection, but Concept Selection further improves the quality of selected labels in Layer 2 through Layer 4, while having the same performance on Layer 1. One potential explanation is due to Layer 1 detecting more limited low level concepts–there is less variance in candidate descriptions identified in Concept Generation (Step 2), resulting in similar ratings across the set of candidate concepts $T$. We can see some individual examples of the improvement Concept Selection provides in Figure 3, with the new labels yielding more specific and accurate descriptions of the neuron. For example Layer 2 Neuron 312 becomes more specific *colorful festive settings* instead of generic *Visual Elements*.

---

[2]Source: https://github.com/first20hours/google-10000-english/blob/master/20k.txt

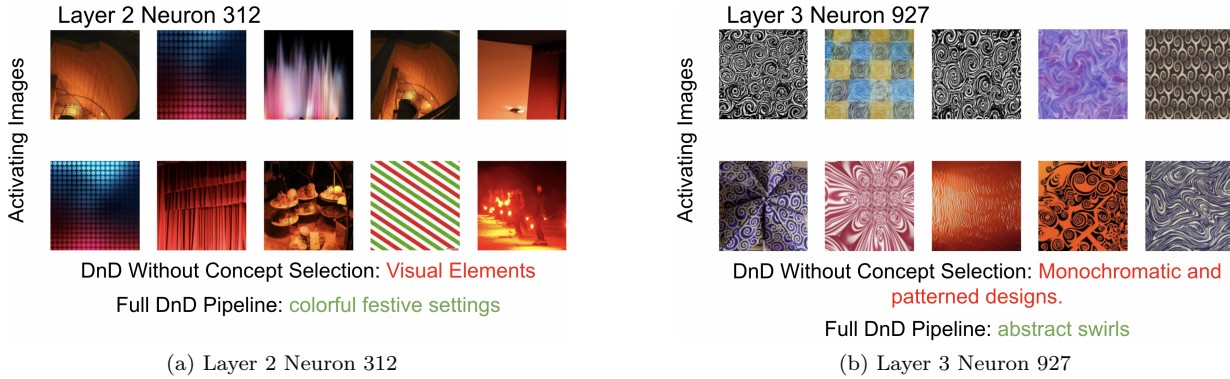

(a) Layer 2 Neuron 312            (b) Layer 3 Neuron 927

Figure 3: **Concept Selection (Step 3) supplements Concept Generation (Step 2) accuracy**. We show that concept selection improves Concept Generation by validating candidate concepts.

Table 5: **Human evaluation results for DnD (w/o Best Concept Selection) versus full Describe-and-Dissect.** Full pipeline improves or maintains performance on every layer in ResNet-50.

| Method / Layer | Layer 1 | Layer 2 | Layer 3 | Layer 4 | **All Layers** |
|---|---|---|---|---|---|
| **DnD** (w/o Best Concept Selection) | **3.54** | 3.77 | 4.00 | 4.02 | 3.84 |
| **DnD** (full pipeline) | **3.54** | **4.00** | **4.24** | **4.13** | **3.97** |

## 5 Use Case: Land-Cover Prediction

One important role of interpretability tools is the ability to create real-world impact. In this section, we study applications in sustainability and climate change by applying **DnD** to the task of land cover prediction–a critical factor for managing water resources, conserving biodiversity, planning sustainable urban development, and mitigating climate change effects. **DnD**'s concept descriptions for neurons not only improve the model performance, but also identify spurious correlations, suggesting a critical role of interpretability techniques in ensuring reliability and building trust in AI systems.

**Setup.** We evaluate our framework on two classification models. **Tile2Vec** (Jean et al., 2019) utilizes a modified ResNet-18 backbone trained to minimize triplet loss between anchor, neighbor, and distant land tiles from the NAIP dataset (Claire Boryan & Craig, 2011). Following Jean et al. (2019), we train a random forest classifier on ResNet-18 embeddings to evaluate accuracy on 27 subclasses of Cropland Data Layer (CDL) labels. We also evaluate a ResNet-50 model trained on labeled EuroSAT images (Helber et al., 2019) with 10 land cover classes. We experiment on two probing datasets. **NAIP** is an aerial imagery dataset updated annually by the USDA. The dataset is composed of crop cover data with annotated ground truth masks from CDL labels. However, due to the fine-grained nature of the 27 subclasses, we also categorize each subclass into six broad superclasses to improve our understanding of the results: **1.** Planted/Cultivated, **2.** Herbaceous/Shrubland, **3.** Urban/Suburban, **4.** Barren, **5.** Forest, **6.** Water/wetlands. We use NAIP for all experiments on the Tile2Vec ResNet-18 model. On EuroSAT ResNet-50, we use the union of EuroSAT $\cup$ ImageNet $\cup$ Broden datasets as the probing dataset.

### 5.1 Locating Conceptual Groupings

We identify neuron clusters detecting similar concepts across Layer 2 of Tile2Vec ResNet-18. For a pair of candidate concepts sets for neurons $n$ and $m$, we define their textual similarity $\mathcal{S}_{n,m}$ as the spatial mean of $L(T_n) \cdot L(T_m)^\top$ where $L(\cdot)$ is the CLIP-ViT-B/16 text encoder. A set of neurons is considered similar if $\mathcal{S}_{n,m} \geq \phi = 0.8$, where $\phi$ controls the minimum similarity threshold between concepts. For practicality, we confine each neuron to exactly one group of highly similar neurons. Low-level concepts are then classified by GPT one-shot classification into the 6 NAIP superclasses.

Figure 4: **Layer 2 Concept Profile.** We cluster neurons with similar concepts and categorize them into 6 NAIP superclasses. Interpretable concepts have more neurons associated with them. Some superclasses do not appear due to dataset bias or intrinsic similarities between classes.

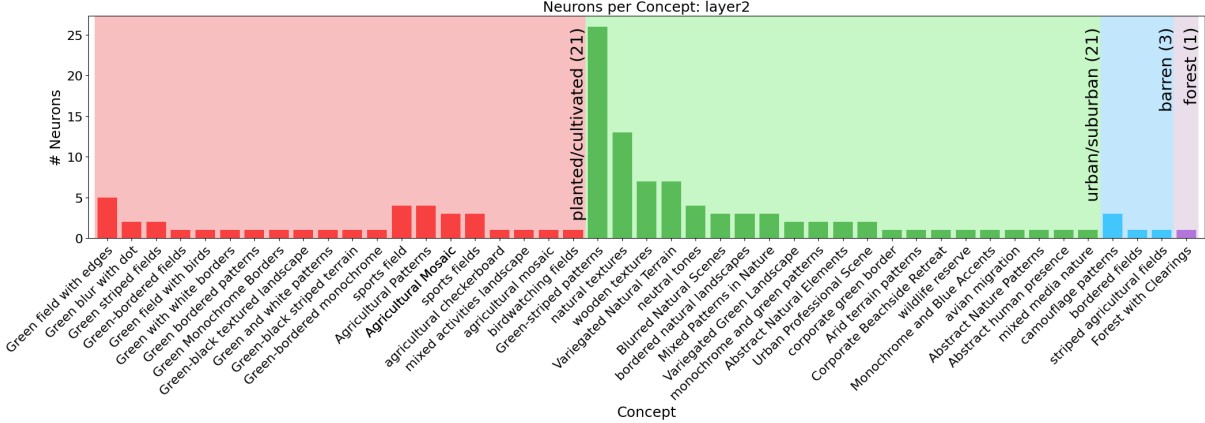

Figure 4 presents concepts from Layer 2, color-coded by superclass. We find that clusters (each bar in Figure 4) containing more neurons are frequently associated with more interpretable features, while clusters with relatively less neurons are related to vague or irrelevant concepts. Based on these insights, we prune neurons with uninterpretable concepts in Section 5.2. We also note that "Herbaceous/Shrubland" and "Water/wetlands" are not associated with concepts in Layer 2. We hypothesize that this is likely due to the intrinsic similarity between the classes and bias in the NAIP dataset. Concepts related to "Herbaceous/Shrubland" are closely related to the "Planted/Cultivated" superclass while "Water/wetlands" images only comprises of ~0.275% of the dataset.

## 5.2   Pruning Uninterpretable Neurons

We conduct a study to identify neurons within Tile2Vec ResNet-18 model that contribute minimally to model accuracy.

Based on Section 5.1, we locate a subset of ungrouped neurons which correlate poorly to other neurons in the network. For our experiment, we define poorly correlating subsets as concepts that activate on only a single neuron. Following the original implementation, we train a classifier on embeddings from a pruned Tile2Vec ResNet-18 model.

Table 6: Pruning uninterpretable neurons in Tile2Vec ResNet18.

| Layer | % Neurons Pruned | Rand. Acc.(%) | Avg. Acc. (%) |
|---|---|---|---|
| No pruning | 0.00 | — | 71.63 |
| All pruned | 100.00 | — | 35.96 |
| Layer 1 | 23.44 | 69.3 | 71.07 |
| Layer 2 | 50.00 | 68.96 | 71.54 |
| Layer 3 | 26.95 | 70.13 | 71.75 |
| Layer 4 | 58.98 | 69.95 | 72.04 |
| Layer 5 | 56.45 | 70.99 | 71.76 |

Table 6 shows model accuracy after identifying and pruning poorly correlating neuron subsets across each layer of the model. Due to high variance in the NAIP dataset, we evaluate the baseline accuracy for a fully pruned network. In this case, we find the prediction is always the "tomatoes" subclass which achieves an accuracy of 35.96% since this subclass comprises of ~35.2% of the data. Our results show that a significant proportion of neurons in the model do not contribute to the overall classification. Particularly in Layers 4 and 5, we are able to prune over 50% of neurons in each layer while achieving a result better than or equal to the baseline performance (no neurons pruned). Across the entire network, the relative small difference in accuracy after pruning suggests human interpretable neurons account for more critical roles within image classification networks compared to uninterpretable neurons. Because we retrain the classifier on the pruned pipeline, the difference in accuracy using DnD pruning and that of random pruning is fairly minor, but we can see random pruning consistently suffers an accuracy drop while DnD pruning does not. In Appendix A.5, we conduct a similar experiment with a fixed classifier.

### 5.3 Characterizing Spurious Correlations

We characterize spurious correlations in the intermediate layers of EuroSAT-trained ResNet-50 by determining common neuron labels through Term Frequency Analysis and studying their relationship to the task. We prune these neurons to further understand the class-wise correlation of spurious concepts, shown in Table 7. Though "fishing" is the most prevalent concept in Layer 4 (41.65% of all neurons), pruning them has no impact on model accuracy. In other words, these fishing neurons are irrelevant to the classification task. "Pink" and "purple" account for 29.74% of Layer 4 neurons and have a much greater impact. These concepts, which are seemingly unrelated to the task, are spuriously correlated to the Forest, Herbaceous Vegetation, Industrial, Pasture, Residential, and Sea-Lake classes. However, the Annual Crop, Highway, and River classes have weaker correlations with these concepts.

Table 7: Pruning concepts from Layer 4 of EuroSAT-trained ResNet-50.

| Concepts pruned | % of neurons pruned | Class-wise Accuracy (%) | | | | | | | | | | Avg. Acc. (%) |
|---|---|---|---|---|---|---|---|---|---|---|---|---|
| | | Ann.-Crop | Forest | Herb-Veget. | Highway | Industrial | Pasture | Perm.-Crop | Residential | River | Sea-Lake | |
| No pruning | 0 | 95.00 | 98.00 | 93.67 | 96.00 | 96.80 | 91.50 | 90.80 | 98.67 | 92.80 | 97.33 | 95.26 |
| Fishing | 41.65 | 95.00 | 98.00 | 93.67 | 96.00 | 96.80 | 91.50 | 90.80 | 98.67 | 92.80 | 97.33 | 95.26 |
| Pink/purple | 29.74 | 88.33 | 0.00 | 0.00 | 62.00 | 0.00 | 0.00 | 18.80 | 0.00 | 76.00 | 0.00 | 24.33 |

## 6 Conclusions

In this paper, we presented Describe-and-Dissect (**DnD**), a novel method for automatically labeling the functionality of deep vision neurons without the need for labeled training data or a provided concept set. We accomplish this through three important steps including probing set augmentation, candidate concept generation through off-the-shelf general purpose models, and best concept selection with carefully designed scoring functions. Through extensive qualitative, quantitative, and use-case analysis, we show that **DnD** outperforms prior work by providing higher-quality neuron descriptions, greater generality and flexibility, and significant potential for social impact.

**Acknowledgments**

This work is supported in part by National Science Foundation (NSF) awards CNS-1730158, ACI-1540112, ACI-1541349, OAC-1826967, OAC-2112167, CNS-2100237, CNS-2120019, the University of California Office of the President, and the University of California San Diego's California Institute for Telecommunications and Information Technology/Qualcomm Institute. Thanks to CENIC for the 100Gbps networks. This work used Expanse CPU, GPU and Storage at SDSC through allocation CIS230152 from the Advanced Cyberinfrastructure Coordination Ecosystem: Services & Support program, which is supported by National Science Foundation grants 2138259, 2138286, 2138307, 2137603, and 2138296. The authors thank REHS program (Research Experience for High School students) in San Diego Supercomputer Center. The authors are partially supported by National Science Foundation under Grant No. 2107189, 2313105, 2430539, Hellman Fellowship, and Intel Rising Star Faculty Award. The authors would also like to thank anonymous reviewers and the editor for valuable feedback to improve the manuscript.

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

# A  Appendix

In the appendix, we discuss specifics of the methods and experiments presented in the paper and provide additional results. Appendix Section A.1 provides further details on the method pipeline, Section A.2 presents additional qualitative evaluations, Section A.3 shows results for quantitative comparison to MILANNOTA-TIONS, Section A.4 provides detailed ablation studies on each step of the **DnD** pipeline, Section A.5 extends our land coverage use case experiment, Section A.6 presents an additional use for **DnD** as an OOD classifier, Section A.7 shows an extension of **DnD** to return multiple concept labels, Section A.8 provides an analysis of the diversity of our method's labels, Section A.9 showcases failure cases of DnD, Section A.10 discusses the limitations and directions for future work, and Section A.11 outlines the broader impact of our work.

**Appendix Outline**

1. **A.1** Method Details
   (a) **A.1.1** Detailed Description of Attention Cropping
   (b) **A.1.2** Holistic vs. Local Concepts
   (c) **A.1.3** GPT Prompt
   (d) **A.1.4** Scoring Function Details
   (e) **A.1.5** Selecting a Scoring Function
   (f) **A.1.6** Amazon Mechanical Turk Setup

2. **A.2** Qualitative Evaluation

3. **A.3** Quantitative Results: MILANNOTATIONS

4. **A.4** Ablation Studies
   (a) **A.4.1** Attention Cropping
   (b) **A.4.2** Image Captioning with Fixed Concept Set
   (c) **A.4.3** Image-to-Text Model
   (d) **A.4.4** Multimodal GPT
   (e) **A.4.5** Effects of Large-Language-Model Choice
   (f) **A.4.6** Effects of GPT Concept Summarization

5. **A.5** Use Case: Fixed Classifier

6. **A.6** Additional Use Case

7. **A.7** Multiple Labels

8. **A.8** Diversity Analysis

9. **A.9** Failure Cases

10. **A.10** Limitations

11. **A.11** Broader Impact

### A.1 Method Details

### A.1.1 Detailed Description of Attention Cropping

The attention cropping algorithm described in Section 3.1 is composed of three primary procedures:

- **Step 1.** Compute the optimal global threshold $\lambda$ for the activation map of a given neuron $n$ using Otsu's Method (Otsu, 1979).

- **Step 2.** Highlight contours of salient regions on the activation map that activate higher than $\lambda$.

- **Step 3.** Crop $\alpha$ largest salient regions from the original activating image that have an IoU score less than $\eta$ with all prior cropped regions.

Here, we provide more implementation details for each step:

- **Step 1. Global Thresholding using Otsu's Method.** For an activation map of neuron $n$, we define regions with high activations as the "foreground" and regions with low activations as the "background". Otsu's Method (Otsu, 1979) automatically calculates threshold values which maximizes interclass variance between foreground and background pixels. Interclass variance $\sigma_B^2$ is defined by $\sigma_B^2 = W_b W_f (\mu_b - \mu_f)^2$, where $W_{b,f}$ denotes weights of background and foreground pixels and $\mu_{b,f}$ denotes the mean intensity of background and foreground pixels respectively. For global thresholding used in **DnD**, $W_f$ is the fraction of pixels in activation map $M_n(x_i)$ above potential threshold $\lambda$ while $W_b$ is the fraction of pixels below $\lambda$. A value of $\lambda$ that maximizes $\sigma_B^2$ is used as the global threshold.

- **Step 2. Highlight Contours of Salient Regions.** **DnD** utilizes OpenCV's contour detection algorithm to identify changes within image color in the binary masked activation map. Traced contours segments are compressed into four end points, where each end point represents a vertex of the bounding box for the salient region.

- **Step 3. Crop Activating Images.** Bounding boxes recorded from **Step 2** are overlaid on corresponding activating images from $D_{probe}$, which are cropped to form $D_{cropped}$. To prevent overlapping attention crops, the IoU score between every crop in $D_{cropped}$ is less than an empirically set parameter $\eta$, where high $\eta$ values yield attention crops with greater similarities. We use $\eta = 4$ across all experiments.

We also visualize these steps in the below Figure 5 for better understanding.

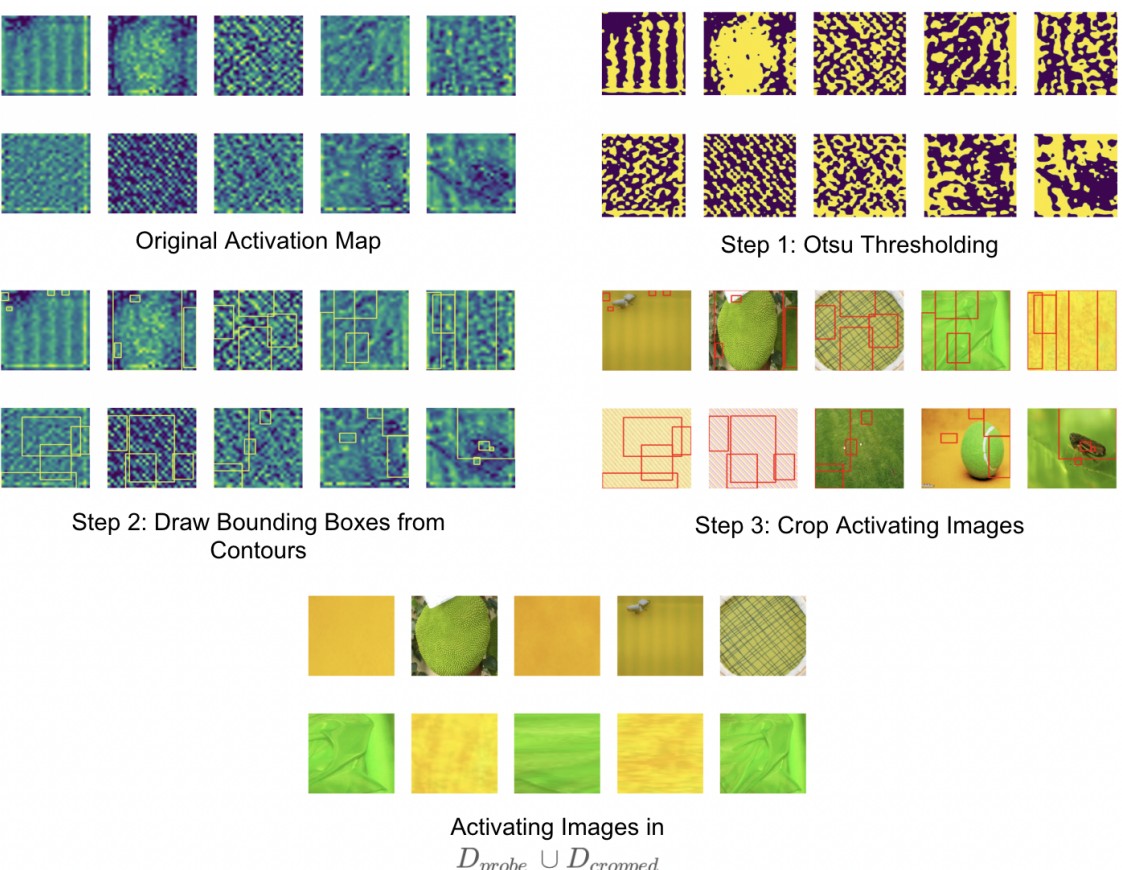

Figure 5: **Detailed Visualization of Attention Cropping Pipeline.** All three steps of attention cropping are shown for Layer 2 Neuron 165. **Steps 1 and 2** illustrate the derivation of bounding boxes from salient regions in the original activation map and are overlaid on original activating images from $\mathcal{D}_{probe}$ in **Step 3**. Cropped images are added back to $\mathcal{D}_{probe}$ to form $\mathcal{D}_{probe} \cup \mathcal{D}_{cropped}$.

### A.1.2 Holistic vs. Local Concepts

To improve the accuracy of **DnD**, we form the probing dataset with cropped images representing local concepts and uncropped images representing holistic concepts. We define local concepts as concepts contained to a portion of the image and holistic concepts as emergent concepts represented by the whole image. This is necessary as solely using image crops can results in vague or spuriously correlated descriptions. We show two examples in Figure 6. In Layer 1 Neuron 64, uncropped images help produce nuanced concepts: cropped images only represent "blue" while uncropped images reflect the "aquatic" concept. Layer 3 Neuron 120 presents an example where cropped images activate on highly localized regions, making the image-to-text model prone to produce spurious captions. Uncropped images (last two images) make the "canine" concept more evident.

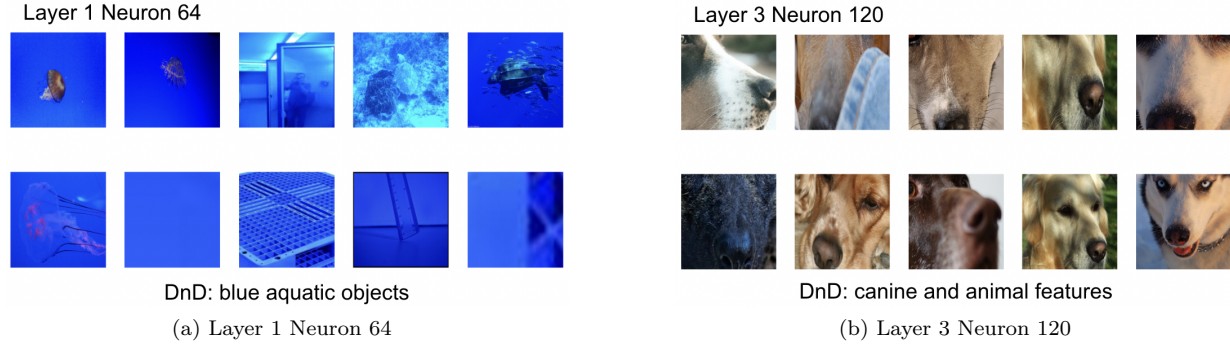

(a) Layer 1 Neuron 64        (b) Layer 3 Neuron 120

Figure 6: **Holistic vs Local Concepts**. We show two cases where a union of local and holistic concepts helps improve model accuracy. Augmenting the dataset reduces vague and spurious captioning.

### A.1.3 GPT Prompt

The Concept Selection process utilizes GPT to identify and coherently summarize similarities between image captions of every image in $I$. Due to the generative nature of **DnD**, engineering a precise prompt is crucial to the method's overall performance. The use of GPT in **DnD** can be split into two primary cases, each with an individual prompt: 1) Summarizing image captions to better bring out the underlying concepts, and 2) Detecting similarities between the image captions. We explain each part separately.

**1. Summarizing Image Captions**. BLIP's image captions frequently contain extraneous details that detract from the ideas represented. To resolve this problem, captions are fed through GPT and summarized into simpler descriptions to represent the concept. We use the following prompt:

> "Only state your answer without a period and quotation marks. Do not number your answer. State one coherent and concise concept label that simplifies the following description and deletes any unnecessary details:"

**2. Detecting Similarities Between Image Captions**. To describe the similarity between image captions, we design the following prompt:

> "Only state your answer without a period and quotation marks and do not simply repeat the descriptions. State one coherent and concise concept label that is 1-5 words long and can semantically summarize and represent most, not necessarily all, of the conceptual similarities in the following descriptions:"

In addition to the prompt, we use few shot-prompting to feed GPT two sets of example descriptions along with the respective human-identified similarities:

**Example 1:**

**Human-Identified Similarity**: "multicolored textiles"
**Image Captions**:

- "a purple background with a very soft texture"

- "a brown background with a diagonal pattern of lines and lines"

- "a white windmill with a red door and a red door in the middle of the picture"

- "a beige background with a rough texture of linen"

- "a beige background with a rough texture and a very soft texture"

**Example 2:**

**Human-Identified Similarity**: "red-themed scenes"
**Image Captions**:

- "a little girl is sitting in a red tractor with the word sofy on the front"

- "a toy car sits on a red ottoman in a play room"

- "a red dress with silver studs and a silver belt"

- "a red chevrolet camaro is on display at a car show"

- "a red spool of a cable with the word red on it"

We also include the prompt used for our Use Case one-shot-classification.

**3.** **Use Case Superclass Classification**. To classify neurons into broader NAIP superclasses we use the following prompt:

> "State one coherent and concise concept label 1-5 words long related to landscapes/satellite imagery that semantically summarizes and represents most, not necessarily all, of the conceptual similarities in the following descriptions. Focus on colors, textures, and patterns. After, print one most likely natural landscape described by the satellite imagery captions, in parenthesis, on the same line. Be confident and do not be vague: "

### A.1.4   Scoring Function Details

Though scoring functions all seek to accomplish the same purpose, the logic behind them can vary greatly and have different characteristics. Simplest scoring functions, such as mean, can be easily skewed by outliers in $\mathcal{H}_j$, resulting in final concepts that are loosely related to the features detected by neurons. In this section, we discuss three different scoring functions and a combination of them which we experimented with in Section A.1.5. The concept with highest score is chosen as the final description for neuron $n$.

- *Mean.* Simply score concept $j$ as the negative mean of its image activation rankings $\mathcal{H}_j$. Concepts where each image activates the neuron highly will receive low ranks and therefore high score. We use the subscript $M$ to denote it's the score using *Mean.*

$$score_M(\mathcal{H}_j) = -\frac{1}{Q}\sum_{i=1}^{Q}\mathcal{H}_{ji}$$

- *TopK Squared.* Given $\mathcal{H}_j$, the score is computed as the mean of the squares of $\beta$ lowest ranking images for the concept. For our experiments, we use $\beta = 5$. This reduces reliance on few poor images. We use the subscript $TK$ to denote the score using *TopK Squared*:

$$score_{TK}(\mathcal{H}_j, \beta) = -\frac{1}{\beta}\sum_{i=1}^{\beta}\mathcal{H}_{ji}^2$$

- *Image Products.* Let set $\mathcal{D}_j^t \subset \mathcal{D}_j$, such that it keeps the $t$ highest activating images from $\mathcal{D}_j$. From the original set of activating images $I$ and $\mathcal{D}_j^t$, the Image Product score is defined as the average cosine similarity between original highly activating images and the generated images for the concept $j$. We measure this similarity using CLIP-ViT-B/16 Radford et al. (2021) as our image encoder $E(\cdot)$:

$$score_{IP}(I, \mathcal{D}_j^t) = \frac{1}{|I| \cdot |\mathcal{D}_j^t|}\sum_{x \in I}\sum_{x^{new} \in \mathcal{D}_j^t}(E(x) \cdot E(x^{new}))$$

  See Figure 7 for an illustration of *Image Product*. Intuitively, *Image Products* selects the candidate concept whose generated images are most similar to the original highly activating images. However, *Image Product* doesn't actually account for how highly the new images activate the target neuron, which is why we chose to use this method to supplement other scoring functions. The enhanced scoring method, *TopK Squared + Image Products*, is described below.

- *TopK Squared + Image Products.* This scoring function uses both image similarity and neuron activations to select the best concept. The method combines *Image Products and TopK Squared* by multiplying the *Image Product* score with the relative rankings of each concept's *TopK Squared* score. We define $\mathcal{R}_{TK} = \{ \, score_{TK}(\mathcal{H}_j, \beta), \, \forall \, j \in \{1, ..., N\}\}$ as the set of *TopK Squared* scores for different descriptions. The final score is then:

$$score_{TK-IP}(\mathcal{H}_j, \beta, I, \mathcal{D}_j^t) = (N - \text{Rank}(score_{TK}(\mathcal{H}_j, \beta); \mathcal{R}_{TK})) \cdot score_{IP}(I, \mathcal{D}_j^t),$$

  where we use $N - \text{Rank}(\cdot)$ to invert the ranks of *TopK Squared* so low ranks result in a high score.

In Section A.1.5, we compare the different functions and note that our model is largely robust to the choice of scoring function with all performing similarly well. We use the *TopK Squared + Image Products* scoring function for all experiments unless otherwise specified.

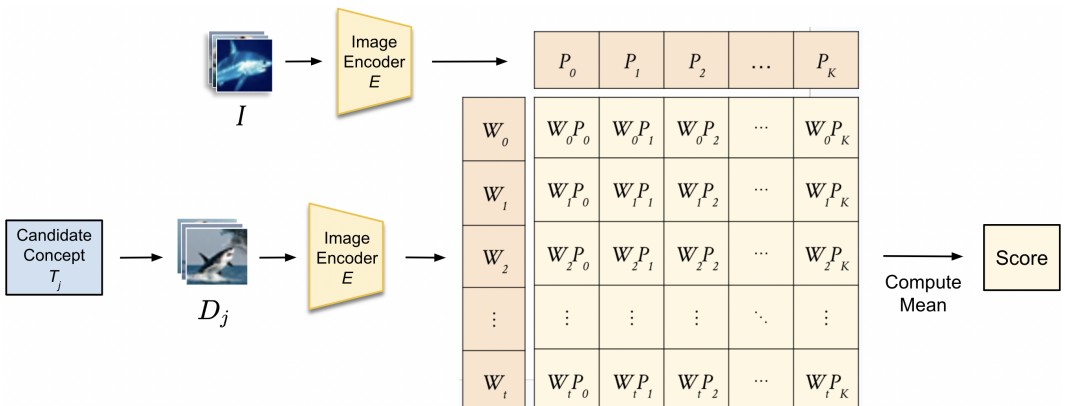

Figure 7: **Image Product Scoring Function.** In the diagram, we let $W_i = E(\mathcal{D}_j^i)$ and $P_i = E(I_i)$. Image Products computes the mean of $W_i \cdot P_i$.

### A.1.5 Selecting a Scoring Function

We again use ResNet-50 with Imagenet ∪ Broden as the probing dataset. 50 neurons were randomly chosen from each of the 4 hidden layers, with each neuron evaluated twice, again rating the quality of descriptions on a scale 1-5. The participants in this experiment were volunteers with no knowledge of which descriptions were provided by which methods. Table 8 shows the performance of different scoring functions described in Section A.1.4. We observe that all three scoring functions presented perform similarly well, and shows our algorithm is robust and not dependent on the choice of scoring function.

Table 8: **Comparison of scoring functions**. Total of 276 evaluations were performed on interpretable neurons (neurons deemed uninterpretable by raters were excluded) from the first 4 layers of ResNet-50 on a scale from 1 to 5. We observe a small margin of difference between the total averages of each scoring function's ratings. Across all 4 layers, the worst and best performing scoring functions differ in average rating by only 0.04 so our algorithm is not dependent on the choice of scoring function.

| Scoring Function / Layer | Layer 1 | Layer 2 | Layer 3 | Layer 4 | All Layers |
| --- | --- | --- | --- | --- | --- |
| Mean | **3.95** | 3.92 | 4.14 | **3.82** | **3.95** |
| TopK Squared | 3.73 | 3.91 | 4.23 | 3.80 | 3.91 |
| TopK Squared + Image Products | 3.77 | **3.94** | **4.28** | 3.78 | 3.93 |

### A.1.6 Amazon Mechanical Turk Setup

In this section we explain the specifics of our Amazon Mechanical Turk experiment setup in Section 4.3. In the interface shown in Figure 8, we display the top 10 highly activating images for a neuron of interest and prompt workers to answer the question: "Does description accurately describe most of the above 10 images?" Each given description is rated on a 1-5 scale with 1 representing "Strongly Disagree", 3 representing "Neither Agree nor Disagree", and 5 representing "Strongly Agree". Evaluators are finally asked to select the label that best describes the set of 10 images.

We used the final question to ensure the validity of participants' assessments by ensuring their choice for best description is consistent with description ratings. If the label selected as best description in the final question was not (one of) the highest rated descriptions by the user, we deem the response invalid. In our analysis we only kept valid responses, which amounted to 1744/2400 total responses for Resnet-50 and 443/600 responses for ResNet-18 (around 75% of the responses were valid).

We collected responses from workers over 18 years old, based in the United States who had $> 98\%$ approval rating and more than 10,000 approved HITs to maximize the quality of the responses. We paid workers \$0.08 per response, and our experiment was deemed exempt by the relevant IRB board.

One downside of our experimental setup is that it is not clear which highly activating images we should display to the users and how to display them, although these choices may be important to the final results. In our experiments, we displayed the images in $\mathcal{D}_{\text{probe}}$ with the highest mean activation as done by CLIP-Dissect (Oikarinen & Weng, 2023), but displaying images with highest max activation would likely be better for MILAN (Hernandez et al., 2022). Unfortunately there is no simple answer to which images we should display that would be good for all methods. See Limitations A.10 for more discussion.

Figure 8: **Amazon Mechanical Turk experiment user interface**. For each neuron, we present participants with the top 10 highly activating images and prompt them to rate each method's label based by how accurately the description represents the set of images. We also ask participants to select the best label from the four descriptions presented.

## A.2  Qualitative Evaluation

Figures 9, 10, 11, and 12 provide supplementary qualitative results for **DnD** along with results produced by baseline methods for ResNet-50 and ResNet-18. Figure 13 showcases our descriptions on random neurons of the ViT-B/16 Vision Transformer trained on ImageNet, showing good descriptions overall and higher quality than those of CLIP-Dissect. Figure 14 extends **DnD** to ResNet-152, demonstrating the ability to scale to larger vision networks. In Figures 15 and 16 we dissect ResNet-50 using the CIFAR-100 training image set as $D_{probe}$, showcasing our method still perfoms well with a different probing dataset. Finally, we present a qualitative comparison with FALCON (Kalibhat et al., 2023) in Figure 17.

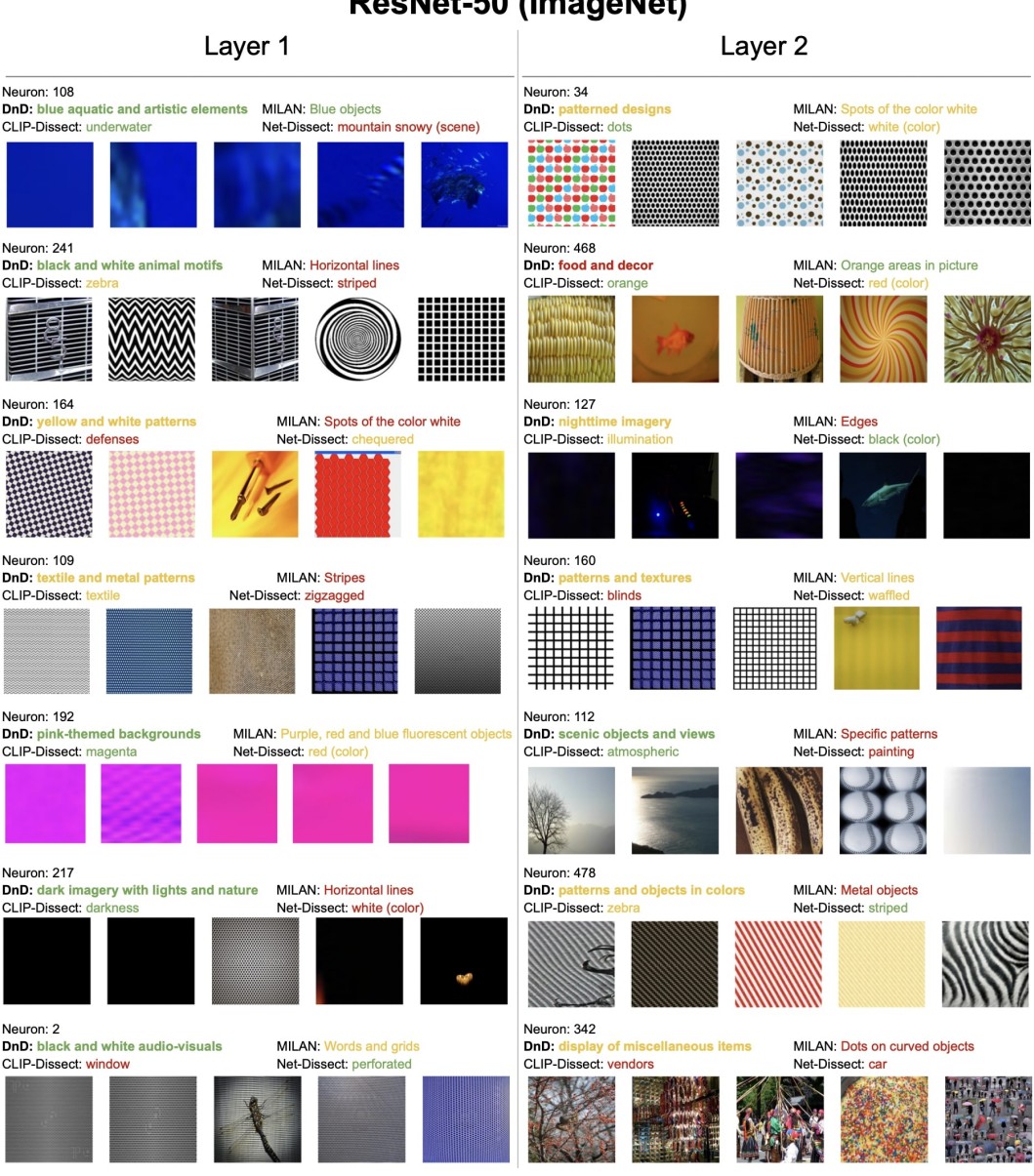

Figure 9: **Additional examples of DnD results from Layer 1 and 2 of ResNet-50**. We showcase a set of randomly selected neurons and their descriptions from Layer 1 and 2 of ResNet-50 trained on ImageNet-1K. Labels are color-coded by whether we believed they were accurate, somewhat correct/vague or imprecise.

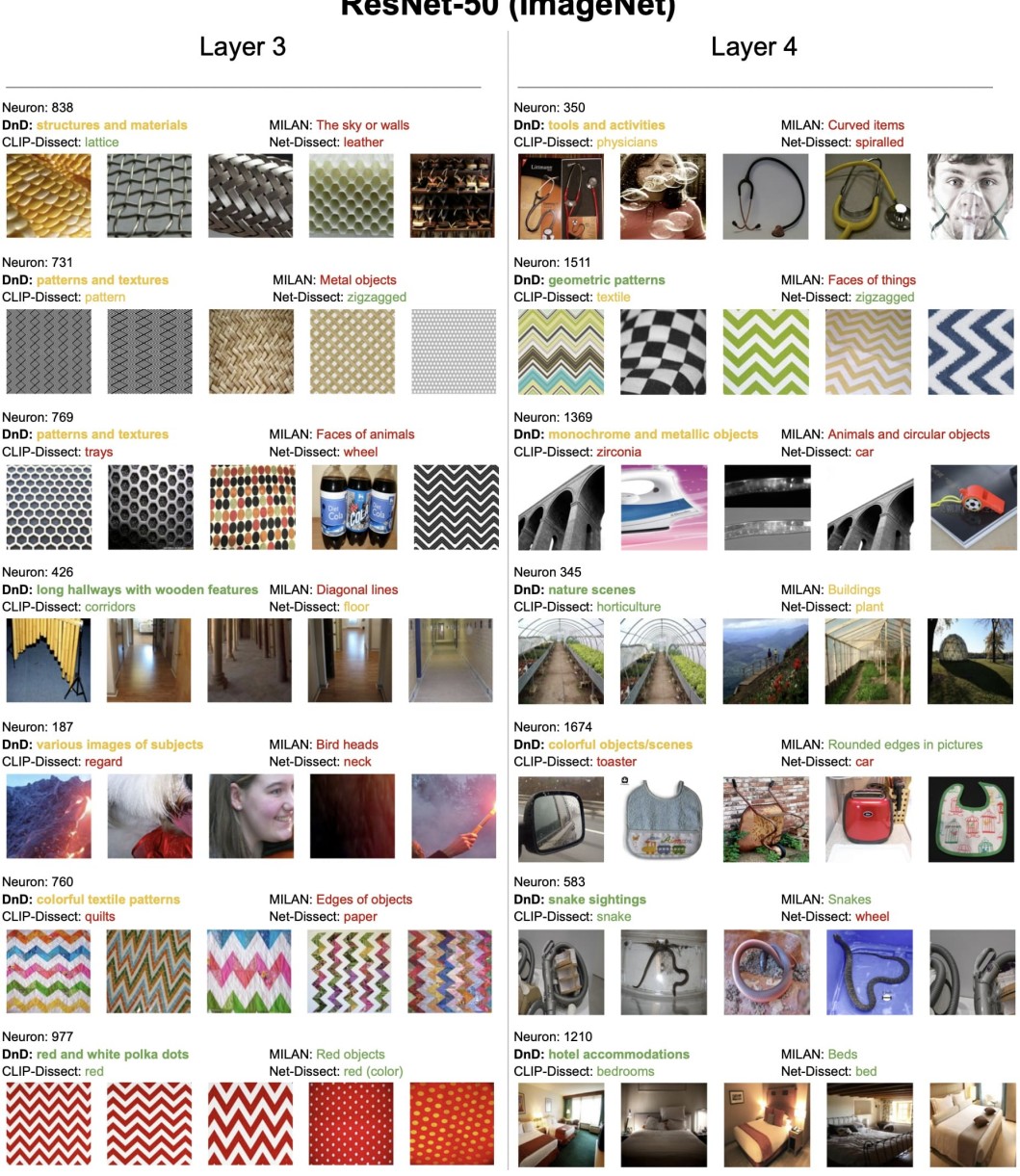

Figure 10: **Additional examples of DnD results from Layer 3 and 4 of ResNet-50**. We showcase a set of randomly selected neurons and their descriptions from Layer 3 and 4 of ResNet-50 trained on ImageNet-1K. Labels are color-coded by whether we believed they were accurate, somewhat correct/vague or imprecise.

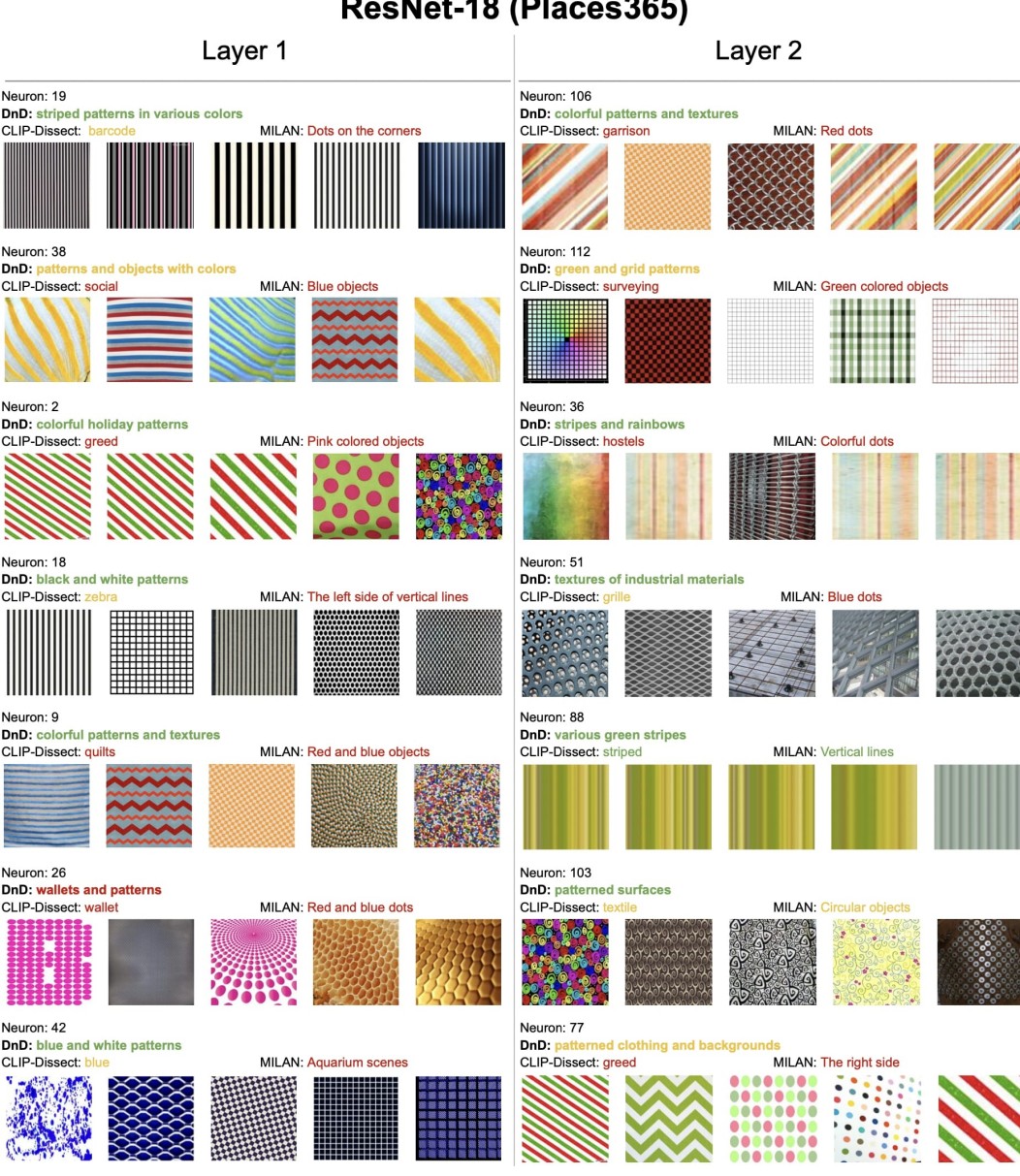

Figure 11: **Additional examples of DnD results from Layer 1 and 2 of ResNet-18**. We showcase a set of randomly selected neurons and their descriptions from Layer 1 and 2 of ResNet-18 trained on Places365. Labels are color-coded by whether we believed they were accurate, somewhat correct/vague or imprecise.

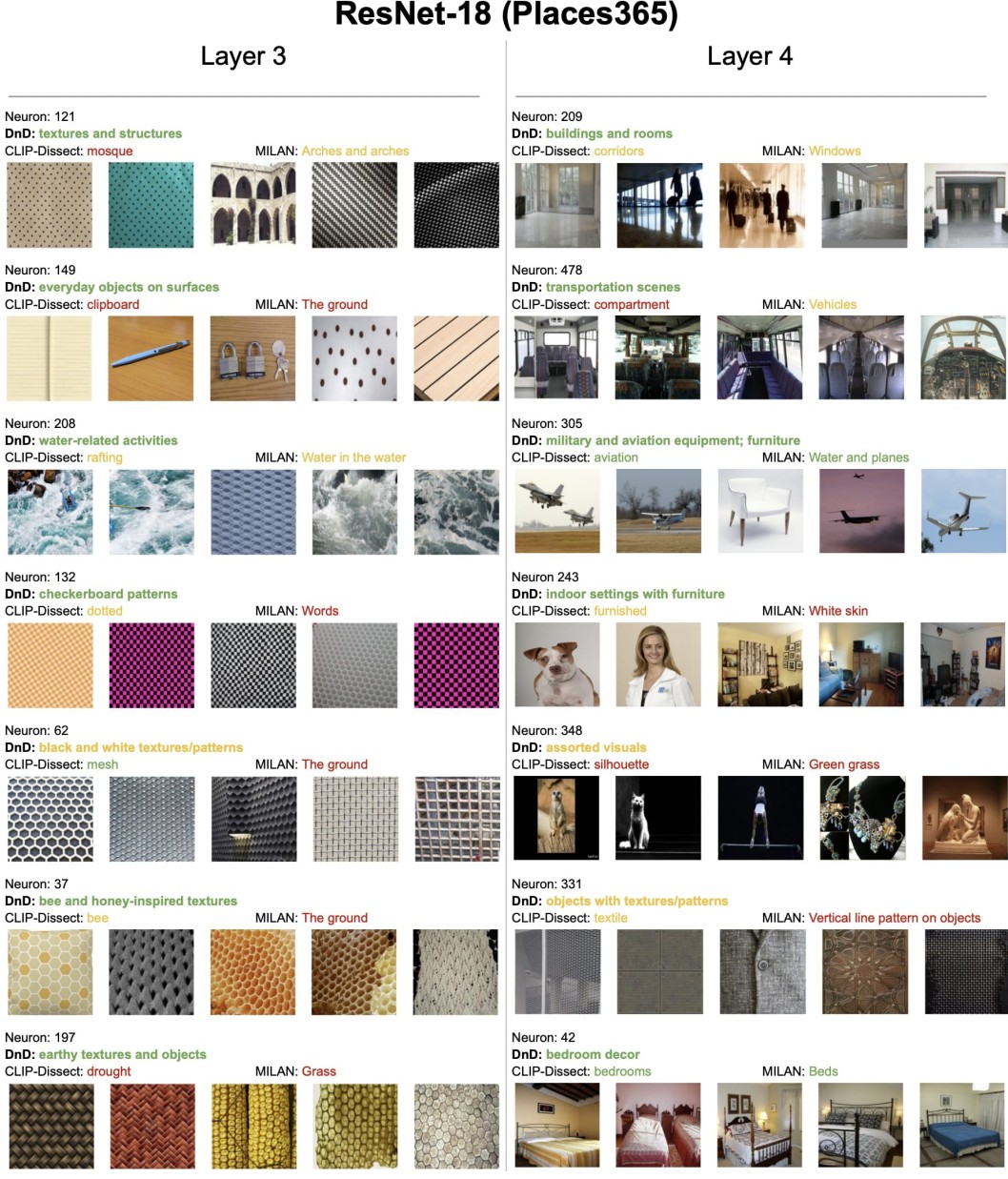

Figure 12: **Additional examples of DnD results from Layer 3 and 4 of ResNet-18**. We showcase a set of randomly selected neurons and their descriptions from Layer 3 and 4 of ResNet-18 trained on Places365. Labels are color-coded by whether we believed they were accurate, somewhat correct/vague or imprecise.

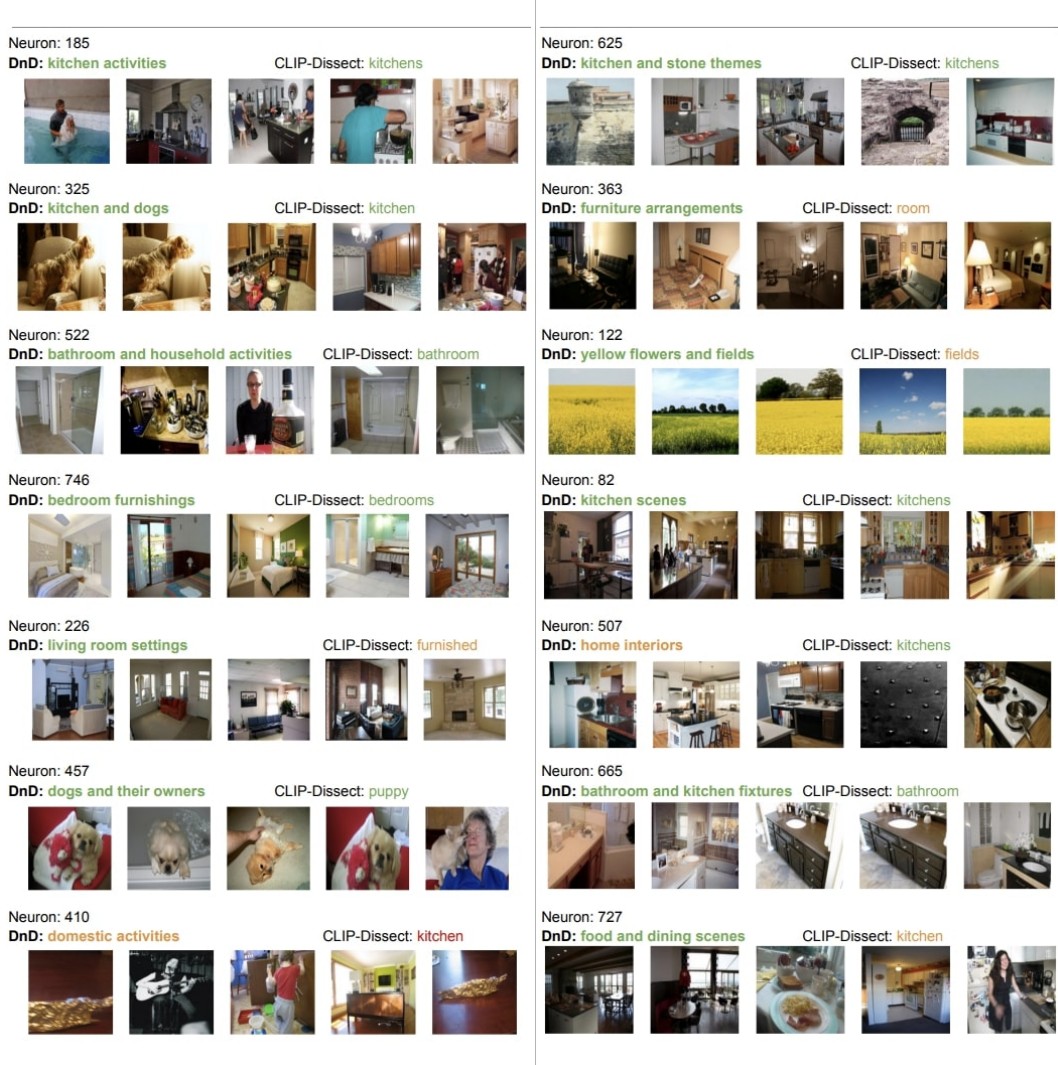

Figure 13: **Examples of DnD results from the encoder layer of ViT-B/16 (ImageNet)**. We showcase a set of randomly selected neurons and their descriptions from the encoder layer of ViT-B/16 trained on ImageNet. **DnD** is model agnostic and can be generalized to any convolutional network. Labels are color-coded by whether we believed they were accurate, somewhat correct/vague or imprecise.

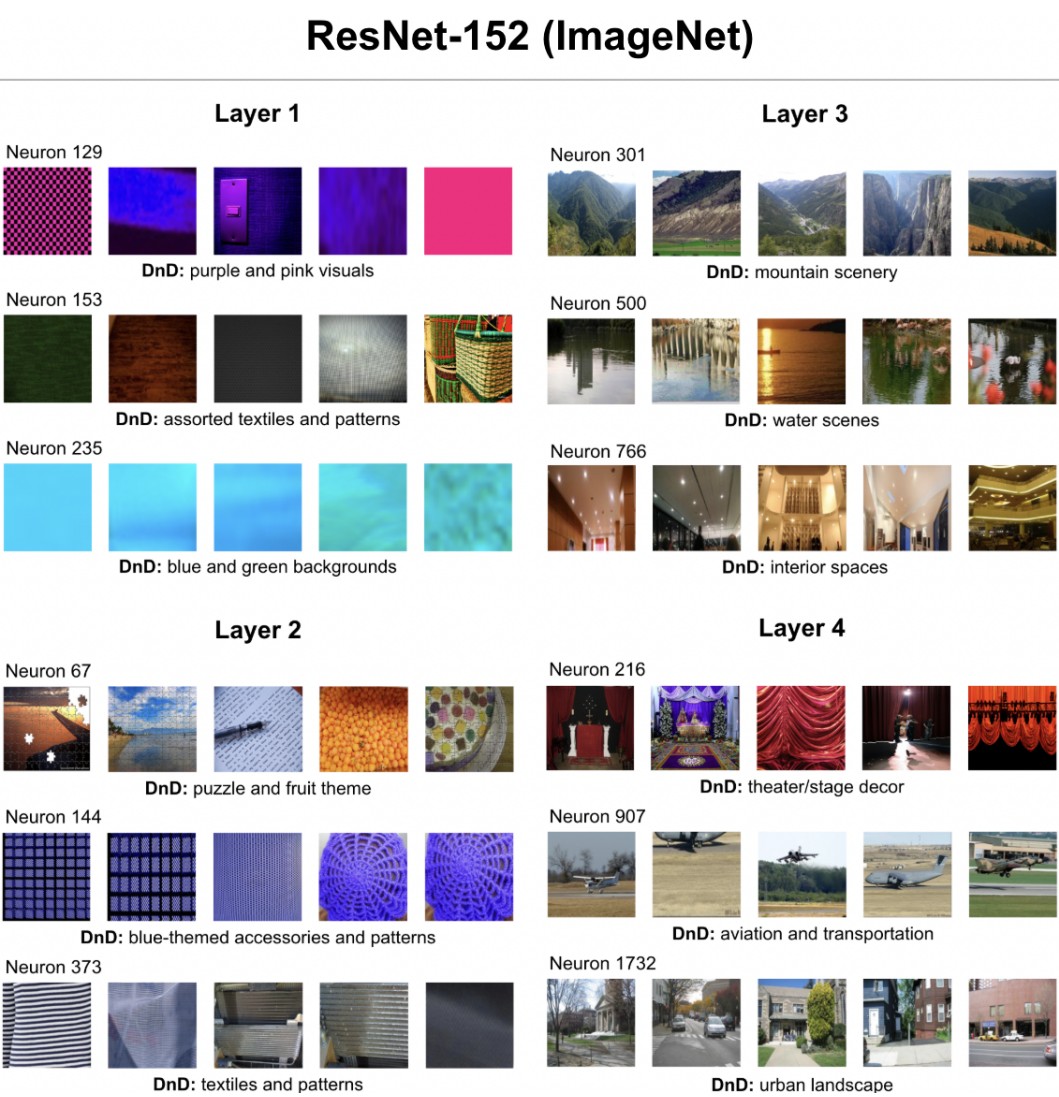

Figure 14: **Examples of DnD results from ResNet-152 (ImageNet)**. We showcase a set of randomly selected neurons and their descriptions across all 4 layers of ResNet-152 trained on ImageNet. **DnD** is scalable to larger vision networks.

Figure 15: **Examples of DnD results from Layer 1 and 2 of ResNet-50 using the training dataset of CIFAR100 as the probing image set**. We showcase a set of randomly selected neurons and their descriptions from Layer 1 and 2 of ResNet-50 trained on ImageNet-1K. **DnD** is probing dataset agnostic and can maintain its high performance on other image sets. Labels are color-coded by whether we believed they were accurate, somewhat correct/vague or imprecise.

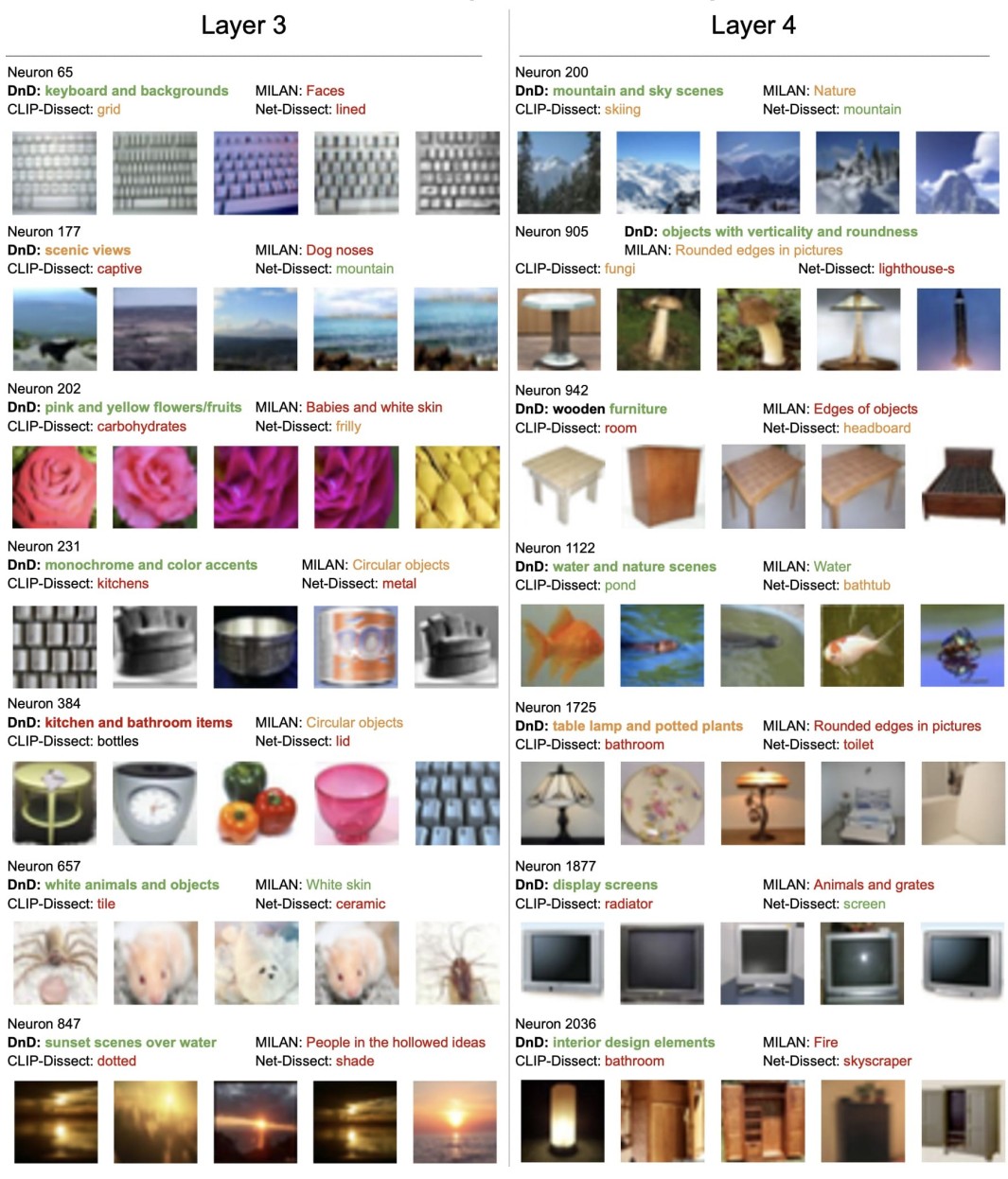

Figure 16: **Examples of DnD results from Layer 3 and 4 of ResNet-50 using the training dataset of CIFAR100 as the probing image set**. We showcase a set of randomly selected neurons and their descriptions from Layer 3 and 4 of ResNet-50 trained on ImageNet-1K. **DnD** is probing dataset agnostic and can maintain its high performance on other image sets. Labels are color-coded by whether we believed they were accurate, somewhat correct/vague or imprecise.

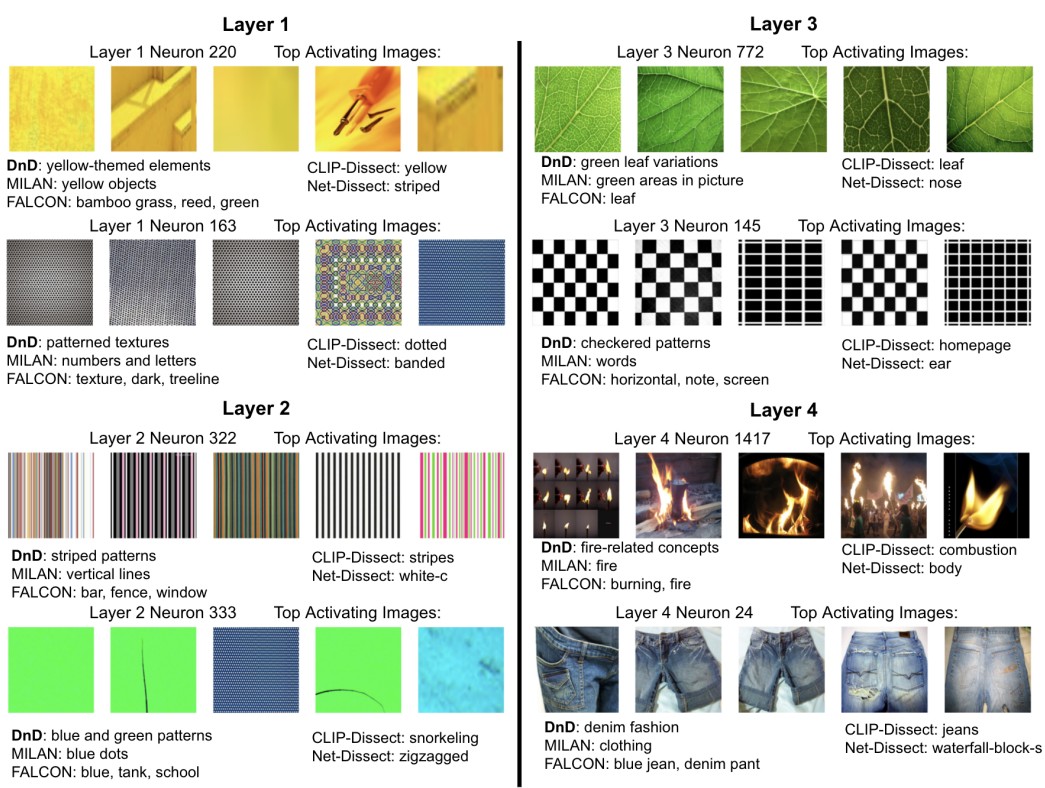

Figure 17: **Examples of DnD results compared with FALCON results from layers across ResNet-50 (ImageNet)**. We showcase a set of randomly selected neurons provided in the appendix of Kalibhat et al. (2023) and their descriptions from layers across ResNet-50 trained on ImageNet-1K. **DnD** performs well compared to FALCON and other contemporary methods.

### A.3 Quantitative Results: MILANNOTATIONS

In this section we discuss using the MILANNOTATIONS dataset (Hernandez et al., 2022) as a set of ground truths to calculate quantitative results on the intermediate layers of ResNet-152. Table 9 displays an experiment done on 103 randomly chosen "reliable" neurons across the 4 intermediate layers of ResNet-152. A neuron was deemed "reliable" if, out of its three corresponding MILANNOTATIONS, two had a CLIP cosine similarity exceeding a certain threshold (for the purposes of this experiment we set that threshold to 0.81). Additionally, we define our summary function $g$ to be spatial max, as this is what was used to calculate the highest activating images for the MILANNOTATIONS. We compare against MILAN trained on Places365 and CLIP-Dissect, using the spatial max summary function for CLIP-Dissect as well. Our results are generally mixed, with **DnD** performing the best (average of 0.7766), then MILAN (average of 0.7698), and finally CLIP-Dissect (average of 0.7390) using CLIP cosine similarity. These results match with BERTScore, as using that metric **DnD** performs the best (average of 0.8489), then MILAN (average of 0.8472), and finally CLIP-Dissect (average of 0.8368). With mpnet cosine similarity, CLIP-Dissect is calculated as the best (average of 0.1970) followed by MILAN (average of 0.1391) and then **DnD** (average of 0.1089). However, the MILANNOTATIONS dataset is very noisy as seen in Figure 18 and thus is largely unreliable for evaluation. The dataset provides three annotations per neuron, each made by a different person. This causes them to often be very different from one another, not representing the correct concept of the neuron. We can see in the table that the average CLIP cosine similarity between the annotations for each of these reliable neurons is 0.7884, the average mpnet cosine similarity is 0.1215, and the average BERTScore is 0.8482. Due to this noisiness, we found that the best performing description is as generic as possible as it will be closest to all descriptions. In fact, we found that a trivial baseline describing each neuron as "depictions" irrespective of the neuron's function outperforms all description methods, including MILAN. This leads us to believe MILANNOTATIONS should not be relied upon to evaluate description quality. Figure 18 showcases descriptions and MILANNOTATIONS for some neurons.

Table 9: **Similarity between predicted labels and "ground truth" MILANNOTATIONS on the 4 intermediate layers of ResNet-152 trained on ImageNet**. We observe that when using CLIP cosine similarity and BERTScore, **DnD** performs the best followed by MILAN and then CLIP-Dissect, but when using MPNet cosine similarity, CLIP-Dissect performs the best followed by MILAN and then **DnD**. Simply labeling every neuron as "depictions" outperforms all other methods, demonstrating the unreliability of MILANNOTATIONS as an evaluation method. The set's noisiness is also shown by the average similarities between the different annotations for each neuron, demonstrating that the annotations are not consistent. We round results to the nearest 4 decimal places, but in instances of ties within a row, we observe further digits and bold accordingly.

| Metric | Layer | Method | | | | Avg Cos Sim Between Annotations |
|---|---|---|---|---|---|---|
| | | MILAN | CLIP-Dissect | Describe-and-Dissect (Ours) | "depiction" | |
| CLIP cos | Layer 1 | 0.7441 | 0.7398 | 0.7454 | **0.7746** | 0.7643 |
| | Layer 2 | 0.7627 | 0.7472 | 0.7872 | **0.7988** | 0.7976 |
| | Layer 3 | 0.7893 | 0.7414 | 0.7858 | **0.7900** | 0.8026 |
| | Layer 4 | 0.7739 | 0.7291 | **0.7825** | 0.7820 | 0.7845 |
| | All Layers | 0.7698 | 0.7390 | 0.7766 | **0.7864** | 0.7884 |
| mpnet cos | Layer 1 | 0.0801 | 0.1962 | 0.0950 | **0.2491** | 0.1124 |
| | Layer 2 | 0.1143 | 0.1850 | 0.1075 | **0.2513** | 0.1167 |
| | Layer 3 | 0.1462 | 0.2039 | 0.1160 | **0.2483** | 0.1279 |
| | Layer 4 | 0.1973 | 0.1995 | 0.1132 | **0.2501** | 0.1252 |
| | All Layers | 0.1391 | 0.1970 | 0.1089 | **0.2496** | 0.1215 |
| BERTScore | Layer 1 | 0.8426 | 0.8282 | 0.8395 | **0.8513** | 0.8392 |
| | Layer 2 | 0.8472 | 0.8348 | 0.8475 | **0.8597** | 0.8467 |
| | Layer 3 | 0.8519 | 0.8438 | 0.8519 | **0.8580** | 0.8490 |
| | Layer 4 | 0.8456 | 0.8374 | **0.8539** | 0.8475 | 0.8501 |
| | All Layers | 0.8472 | 0.8368 | 0.8489 | **0.8541** | 0.8482 |

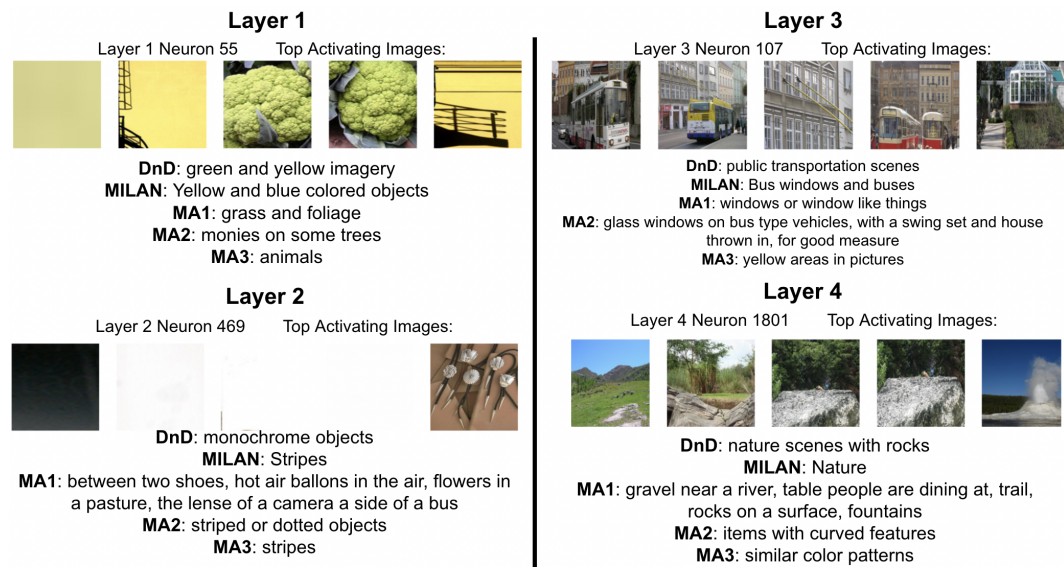

Figure 18: **MILANNOTATIONS from random neurons in ResNet-152 (ImageNet)**. We showcase a set of randomly selected neurons alongside their corresponding **DnD** label, MILAN label, and three MILANNOTATIONS labels (MA). We can see that the three MILANNOTATIONS for each neuron often don't match up well and don't accurately describe the neuron.

### A.4   Ablation Studies

### A.4.1   Ablation: Attention Cropping

Attention cropping is a critical component of **DnD** due to generative image-to-text captioning. Unlike models that utilize fixed concept sets such as CLIP-dissect (Oikarinen & Weng, 2023), image captioning models are prone to spuriously correlated concepts which are largely irrelevant to a neuron's activation. To determine the effects of attention cropping on subsequent processes in the **DnD** pipeline, we evaluate **DnD** on $\mathcal{D}_{probe}$ without augmented image crops from $\mathcal{D}_{cropped}$. We show qualitative examples of this effect in Figure 19.

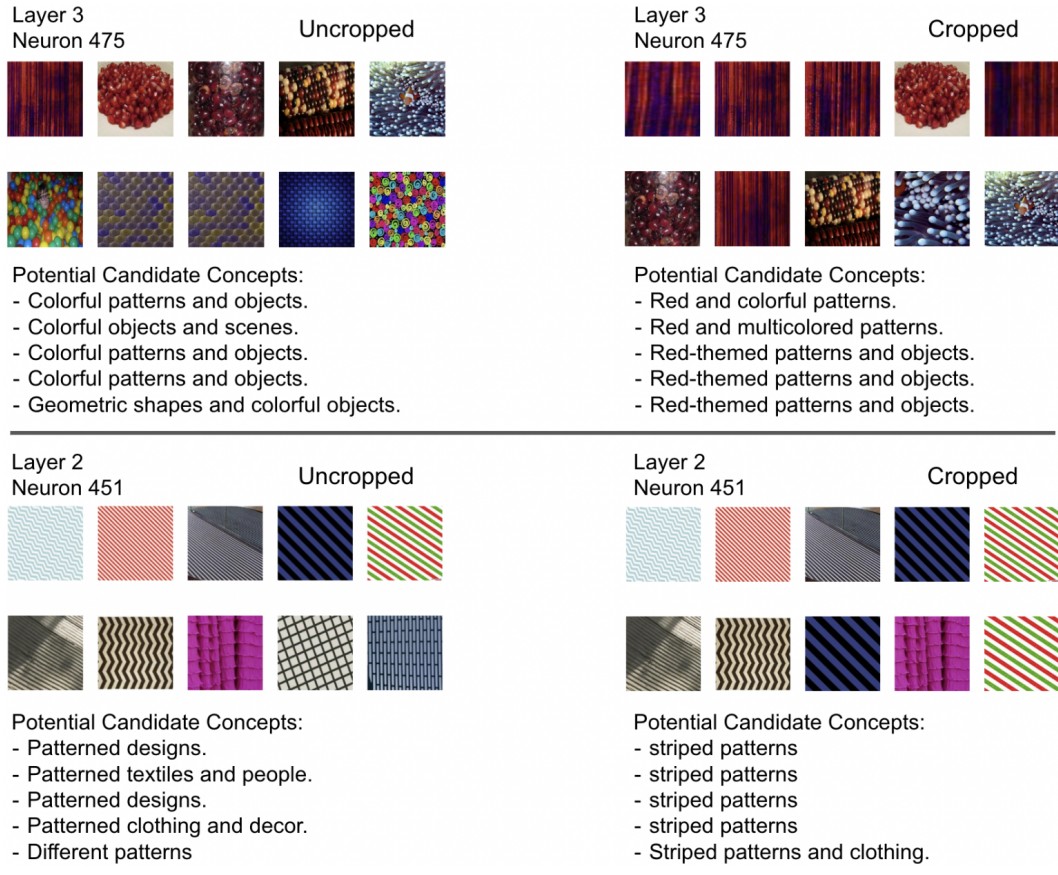

Figure 19: **Examples where Attention Cropping Improves DnD.** The left panel shows the result without attention cropping (**Uncropped**) while the right panel show the result with attention cropping (**Cropped**). It can be seen that Attention cropping eliminates spurious features in highly activating images and improves accuracy of potential candidate concepts.

### A.4.2 Ablation: Image Captioning with Fixed Concept Set

In Section 4.4.1, we explored using fixed concept sets with CLIP (Radford et al., 2021) to generate descriptions for highly activating images rather than BLIP. We present qualitative examples of fixed concept captioning below in Figure 20. Additionally, we also note particular failure cases caused by a lack of expressiveness encapsulated in static concepts, as demonstrated in Figure 21.

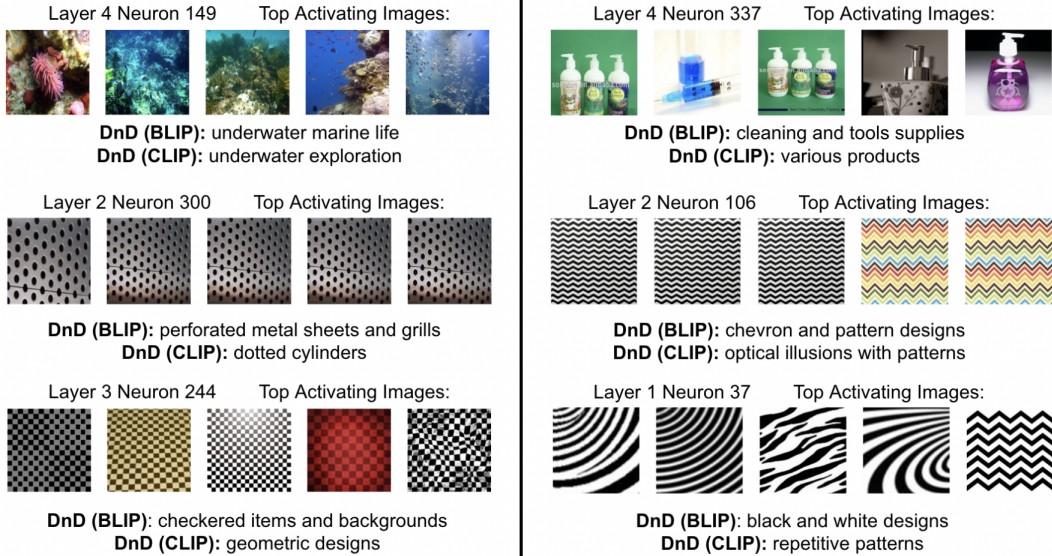

Figure 20: **Examples of DnD with CLIP Image Captioning Compared to DnD (with BLIP).** Despite producing similar results on the FC layer, we see that **DnD** (with BLIP) outperforms **DnD** with CLIP image captioning, especially on intermediate layer neurons. Single word captions fail to fully encapsulate concepts expressed in these layers, resulting in poor **DnD** performance.

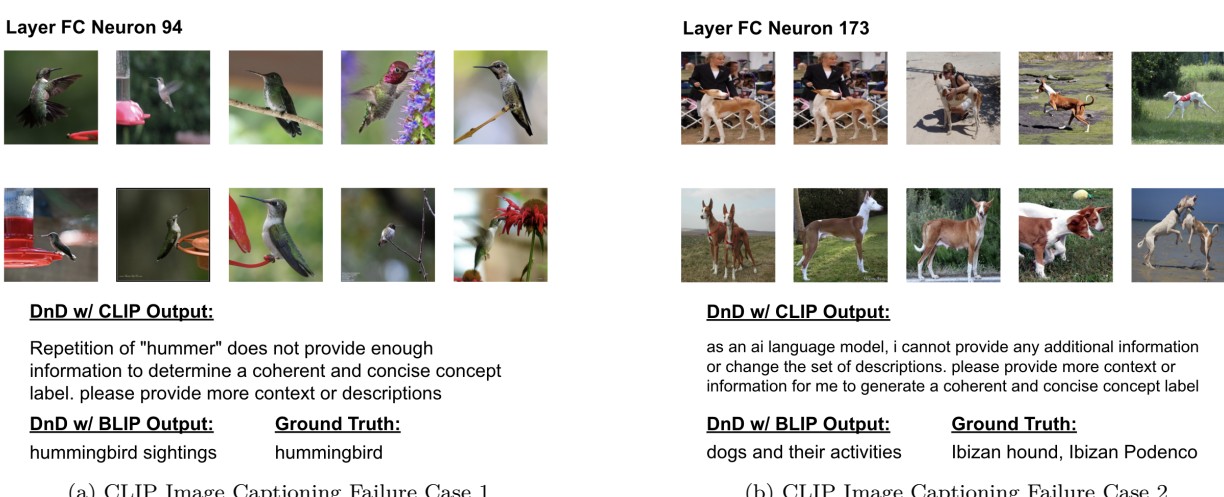

(a) CLIP Image Captioning Failure Case 1

(b) CLIP Image Captioning Failure Case 2

Figure 21: **Failure Cases of CLIP Image Captioning.** Due to the lack of expressiveness of static concept sets, GPT summarization fails to identify conceptual similarities within CLIP image captions. With dynamic concept sets generated by BLIP, the issue is largely avoided.

### A.4.3 Ablation: Image-to-Text Model

In light of advancements in image-to-text models, we compare BLIP (Li et al., 2022) to a more recently released model, BLIP-2 (Li et al., 2023). Unlike BLIP, BLIP-2 leverages frozen image encoders and LLMs while introducing Querying Transformer to bridge the modality gap between two models. We experiment with using BLIP-2 as the image-to-text model and quantitatively compare with BLIP by computing the mean cosine similarity between the best concept chosen from Concept Selection. As discussed in the Appendix A.3, CLIP-ViT-B/16 cosine similarity is a stronger indicator of conceptual connections than all-mpnet-base-v2 similarity for generative labels. Accordingly, CLIP-ViT-B/16 cosine similarity is used as the comparison metric. We evaluate on 50 randomly chosen neurons from each intermediate layer of ResNet-50 and results of the experiment are detailed in Table 10. From Table 10, BLIP and BLIP-2 produce highly related concepts across all four layers of ResNet-50 and a 87.0% similarity across the entire network.

Table 10: **Mean Cosine Similarity Between BLIP and BLIP-2 Labels.** For each layer in RN50, we compute the mean CLIP cosine similarity and BERTScore between BLIP and BLIP-2 labels for 50 randomly chosen neurons. Similar conceptual ideas between both models are reflected in the high similarity scores.

| Metric | Layer 1 | Layer 2 | Layer 3 | Layer 4 | All Layers |
|---|---|---|---|---|---|
| CLIP cos | 0.864 | 0.848 | 0.875 | 0.891 | 0.870 |
| BERTScore | 0.883 | 0.880 | 0.894 | 0.891 | 0.887 |

We also show examples of similar description produced in Figure 22. Somewhat counter-intuitively, our qualitative analysis of neurons in preliminary layers of ResNet-50 reveals examples where BLIP-2 fails to detect low level concepts that BLIP is able to capture. Such failure cases limit **DnD** accuracy by generating inaccurate image captions and adversely affecting the ability of GPT summarization to identify conceptual similarities between captions. We show two such examples in below Figure 23. In general this experiment shows that our pipeline is not very sensitive to the choice of image-captioning model.

**Layer 2 Neuron 25**

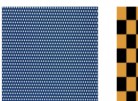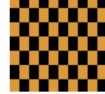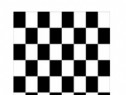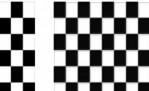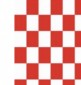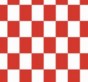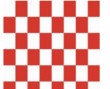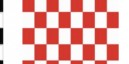

Best BLIP Concept: checkerboard patterns

Best BLIP2 Concept: Checkerboard patterns.

BLIP2 Potential Candidate Concepts

- Checkerboard patterns.
- Checkerboard patterns.
- Checkered and Polka Dot Patterns.
- Checkerboard patterns and beach scenes.
- Checkerboard patterns.

(a) BLIP-2 Similar Example 1

**Layer 4 Neuron 600**

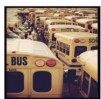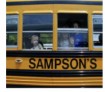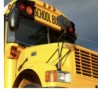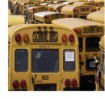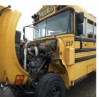

Best BLIP Concept: school buses

Best BLIP2 Concept: school bus transportation

BLIP2 Potential Candidate Concepts

- School bus transportation.
- school bus transportation
- School bus transportation.
- School bus transportation.
- School bus transportation.

(b) BLIP-2 Similar Example 2

Figure 22: **Examples of Similar BLIP and BLIP-2 Labels.** BLIP and BLIP-2 detect similar concepts across all layers of ResNet-50. Two such examples are detailed above.

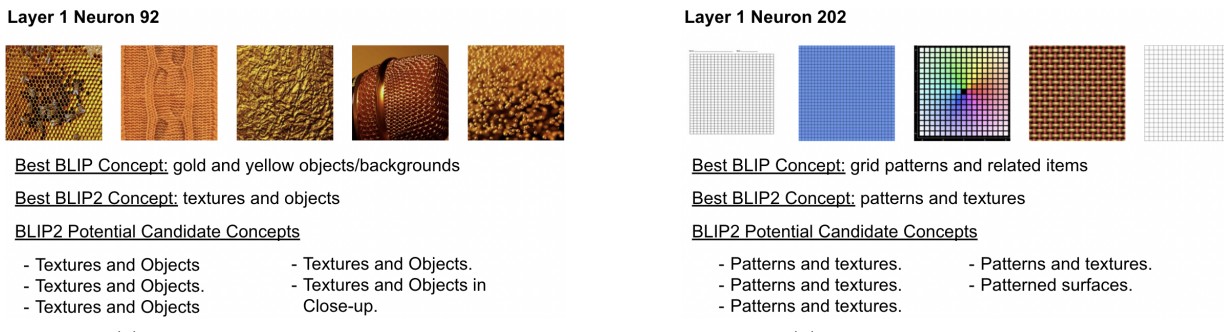

(a) BLIP-2 Failure Case Example 1         (b) BLIP-2 Failure Case Example 2

Figure 23: **Examples of BLIP-2 failure cases.** BLIP-2 overlooks crucial low level concepts in early layers of ResNet-50. Potential Candidate Concepts generated are vague and spuriously correlated, yielding poor results in the final **DnD** label.

### A.4.4  Ablation: Multimodal GPT

To justify using two large models instead of one during Step 2 in our pipeline, we conduct an additional experiment by exchanging BLIP + GPT in Step 2 of **DnD** with the image caption capabilities of GPT-4o. Results in Figure 24 show that due to the large overlap of the GPT models, performance is similar in many examples. However, the original **DnD** pipeline seems to be better than **DnD** w/ GPT-4o when BLIP's image captioning capabilities outperform that of GPT-4o. Though including two foundation models in Step 2 may reduce the flexibility of our pipeline, it can increase performance by specifying fine-tuned models for each designated task.

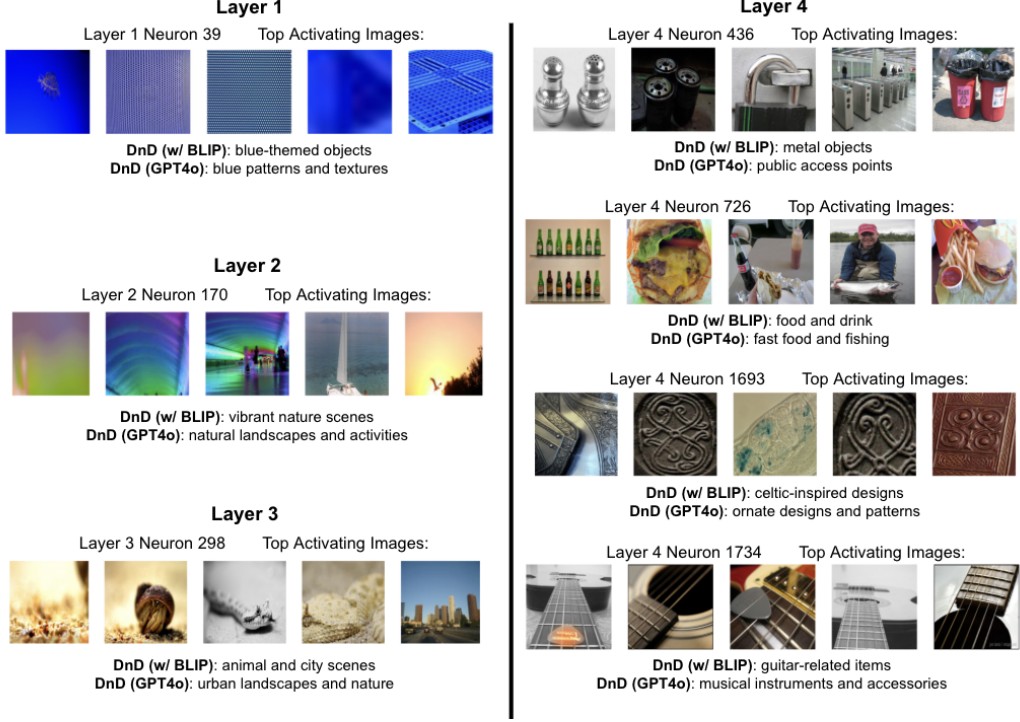

Figure 24: **Neuron examples of GPT-4o replacing BLIP and GPT-3.5-Turbo in the DnD pipeline.**

### A.4.5 Ablation: Effects of Large-Language-Model Choice

To test the robustness of our pipeline at the concept summarization stage, we swap out GPT-3.5 Turbo for GPT-4 Turbo and LLaMA2 (Touvron et al., 2023). For reference, our paper utilizes GPT-3.5 Turbo for all of its primary experiments rather than GPT-4 Turbo because GPT-4 Turbo has 20x the cost of GPT-3.5 Turbo. At the time of developing our pipeline, GPT-3.5 Turbo was the most advanced GPT model provided by OpenAI. LLaMA2, although cost free, is restricted to the public and requires specialized access for use in research.

Our experiment follows the precedence of the setups of our previous experiments in our paper and of the setups utilized in prior works. We evaluate 26 neurons on the FC Layer of ResNet-50 and compare to the ground truth labels provided by the classes. In this experiment, we can see that utilizing other models actually significantly increases **DnD**'s performance in all metrics. As shown in Table 11, not only is **DnD** able to keep up its performance with other models but also improve with new and more advanced ones.

Table 11: **Comparison of LLMs on label quality.** Our experiment shows that GPT-4 Turbo and LLaMA2 significantly increase the cosine similarity score between **DnD** labels and ground truth classes on FC layer neurons, suggesting the ability for our framework to incorporate new advancements in LLMs.

| Metric | **DnD** | **DnD** w/ GPT-4 Turbo | **DnD** w/ LLaMA2 |
|---|---|---|---|
| CLIP cos | 0.6377 | **0.7607** | 0.7407 |
| mpnet cos | 0.1403 | 0.4506 | **0.4726** |
| BERT Score | 0.8172 | **0.8340** | 0.8105 |

### A.4.6    Ablation: Effects of GPT Concept Summarization

**DnD** utilizes OpenAI's GPT-3.5 Turbo to summarize similarities between the image caption generated for each $K$ highly activating images of neuron $n$. This process is a crucial component to improve accuracy and understandability of our explanations, as shown in the ablation studies below.

As mentioned in Appendix A.1.3, GPT summarization is composed of two functionalities: **1.** simplifying image captions and **2.** detecting similarities between captions. Ablating away GPT, we substitute the summarized candidate concept set $T$ with the image captions directly generated by BLIP for the $K$ activating images of neuron $n$. We note two primary drawbacks:

- **Poor Concept Quality.** An important aspect of GPT summarization is the ability to abbreviate nuanced captions into semantically coherent concepts (function **1.**). In particular, a failure case for both BLIP and BLIP-2, shown in Figure 25a, is the repetition of nonsensical elements within image captions. Processing these captions are computationally inefficient and can substantially extend runtime. **DnD** resolves this problem by preprocessing image captions prior to similarity detection.

- **Concept specific to a single image.** Without the summarization step, concepts only describe a single image and fail to capture the shared concept across images. We can see for example that the neuron in Figure 25b is described as a "pile of oranges", which only applies to the fourth image and misses the overall theme of orange color.

**Layer 1 Neuron 155**

Best BLIP Concept: black and white che che…

Potential Candidate Concepts

- perfored perfored sheet perfored perfored…
- black and white che che…
- a baby bi bi bi bi…
- a bi bi bi bi…
- perfored sheet perfored sheet perfored sheet…

(a) GPT Ablation: Poor Quality Concept

**Layer 2 Neuron 418**

Best BLIP Concept: a pile of oranges

Potential Candidate Concepts

- a bunch of colorful colored wires
- a close up of a light on a table
- a spider web in the middle of a spider web
- a pile of oranges
- a person wearing red socks and holding a cell

(b) GPT Ablation: Concept Specific to Single Image

Figure 25: **Failure Cases in GPT-Ablation Study.** Figure (a) illustrates potential failure cases of **DnD** without GPT summarization. BLIP produces illogical image captions that hinder overall interpretability. Figure (b) shows an additional failure case where the omission of concept summarization causes individual image captions to generalize to the entire neuron.

### A.5 Use Case: Fixed Classifier

The Tile2Vec model evaluated contains two parts: 1. A ResNet model to compute embeddings of land tiles, 2. A random forest classifier to predict the land coverage of the output embedding. We use the originally proposed Tile2Vec pipeline by first pruning neurons in the ResNet model and then training the classifier on embeddings of the pruned model. Because we retrain the classifier on the pruned pipeline, the difference in accuracy using DnD pruning and that of random pruning is fairly minor. In this section, we instead evaluate the Tile2Vec model using a fixed classifier. We train the classifier using embeddings from the full unpruned pipeline and evaluate by fixing the classifier while pruning the ResNet model only. We observed a moderate (13.63%) increase in speed after performing pruning on the first layer of Tile2Vec ResNet-18 and further gains are possible if running inference more optimized for sparse models. Table 12 shows results using a similar percentage of pruned neurons as Table 6.

Table 12: **Pruning uninterpretable neurons in Tile2Vec ResNet18.** Due to GPT's generative captioning, the similarity between neuron labels can vary slightly between runs, causing slightly different number of neurons pruned.

| Layer | % of Neurons Pruned | Random Pruning Acc. (%) | Avg. Acc. (%) |
|---|---|---|---|
| No pruning | 0.00 | — | 71.63 |
| All pruned | 100.00 | — | 35.96 |
| Layer 1 | 23.44 | 45.0 | 51.0 |
| Layer 2 | 50.0 | 47.0 | 49.5 |
| Layer 3 | 26.95 | 66.6 | 70.5 |
| Layer 4 | 50.78 | 49.5 | 56.0 |
| Layer 5 | 56.64 | 56.0 | 60.5 |

## A.6    Additional Use Case

To showcase a potential use case for neuron descriptions (and provide another way to quantitatively compare explanation methods), we experimented with using neuron descriptions to find a good classifier for a class missing from the training set. Our setup was as follows: we explained all neurons in Layer 4 of ResNet-50 (ImageNet) using different methods. We then wanted to find neurons in this layer that could serve as the best classifiers for an unseen class, specifically the classes in CIFAR-10, CIFAR-100, and Places365 datasets. Although there is some overlap, these classes are typically much more broad than ImageNet classes. To find a neuron to serve as a classifier, we locate the neuron whose description is closest to the CIFAR/Places365 class name in a text embedding space. We then measure how well that neuron (its average activation) performs as a single class classifier on the respective validation dataset, measured by area under ROC curve. For cases where multiple neurons share the closest description, we average the performance of all neurons with that description.

Results are shown in Table 13. We can see **DnD** performs quite well, reaching AUROC values around 0.74, while MILAN performs much worse. We believe this may be because MILAN descriptions are very generic (likely caused by noisy dataset as discussed in Section 4.3.1), making it hard to find a classifier for a specific class. We believe this is a good measure of explanation quality, as different methods are dissecting the same network, and even if no neurons exist that can directly detect a class, a better method should find a closer approximation.

|  | MILAN | **DnD (Ours)** |
|---|---|---|
| CIFAR10 | 0.5906 | **0.7036** |
| CIFAR100 | 0.6514 | **0.7396** |
| Places365 | 0.650 | **0.709** |

Table 13: The average classification AUC on out of distribution dataset when using neurons with similar description as a classifier. We can see that our **DnD** clearly outperforms MILAN, the only other generative description method.

## A.7 Multiple Labels

As discussed in the Appendix A.10, many neurons in intermediate layers can be considered "polysemantic", meaning that they encode multiple unrelated concepts. Our primary pipeline only produces one label, and though this label can encapsulate more than one concept, providing multiple labels can better account for issues of polysemanticity. We accomplish this by taking the top candidate concepts selected by Concept Selection as our final labels. If the final labels have a CLIP cosine similarity exceeding 0.81, we only take the top labels and eliminate the others from the set of final labels. This allows use to bring out accurate labels that account for distinct concepts prevalent in the neuron. We can see this in Figure 26. For Neuron 508 from Layer 2, **DnD** captures not only the polka dot and metal texture concepts of the neuron, but adds that the textures are primarily black and white. For Neuron 511 from Layer 3, **DnD** labels the neuron as detecting not only interior elements, but also specifying that these elements are mostly black and white.

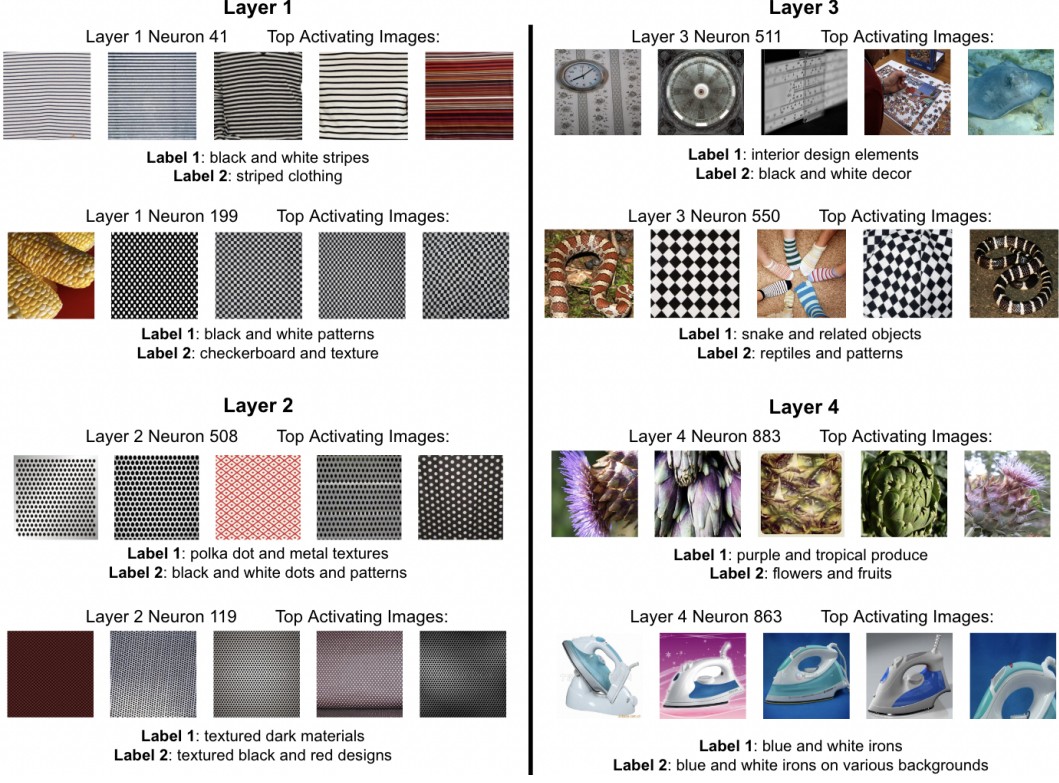

Figure 26: **Example results on ResNet-50 (ImageNet) from the DnD pipeline being modified to produce multiple labels**. We showcase a set of randomly selected neurons to exemplify **DnD**'s capability to provide multiple descriptions. We can see that though some of these neurons can't be described by just a singular label, **DnD**'s multiple labels can describe their various aspects.

### A.8  Diversity Analysis

To more deeply understand the effectiveness and outcomes of our method, we perform an experiment analyzing the relationship between similar highly activating images and similar labels. We follow the setup in Section 4.2. Over the entirety of the FC Layer of ResNet-50 (1000 neurons), we took random pairs of neurons from those 1000, making 500 pairs in total. We then calculated the similarity between each pair's top 10 activating images and calculated the CLIP cosine similarity between each pair's labels outputted by **DnD**. Finally, we calculated the correlation coefficient between image similarity and label similarity with each pair being a datapoint and found that value to be 0.4406. We can see there is a positive correlation between similar highly activating images and similar **DnD** labels. In general, **DnD** provides similar labels for neurons with similar highly activating images while more varying labels for neurons with differing highly activating images.

### A.9 Failure Cases

In this section we analyze a few examples in Figure 27 that can cause **DnD** to produce suboptimal neuron labels. Neuron 106 from Layer 3 exemplifies how **DnD** can occasionally focus on one specific element from the highly activating image set when the true concept is less obvious. Neuron 193 from Layer 3 shows how when given an uninterpretable or polysemantic neuron, **DnD** defaults to a generic label like "patterned items." These neurons are further discussed in sections A.7 and A.10. Neuron 515 from Layer 3 demonstrates how **DnD** may produce a more generic and altered version of the true concept when the true concept is more specific. We can see that the true concept represented by the neuron is likely something along the lines of "animal ears," but **DnD** is not able to identify the ears and adds in an incorrect portion of "resting" to the label. Finally, neuron 1880 from Layer 4 showcases how it's hard for **DnD** to detect more spacial and abstract representations of the concept. We can see that the true concept should be likely along the lines of "line patterns" but since the lines are embedded into other concepts like nature elements, **DnD** isn't able to detect it.

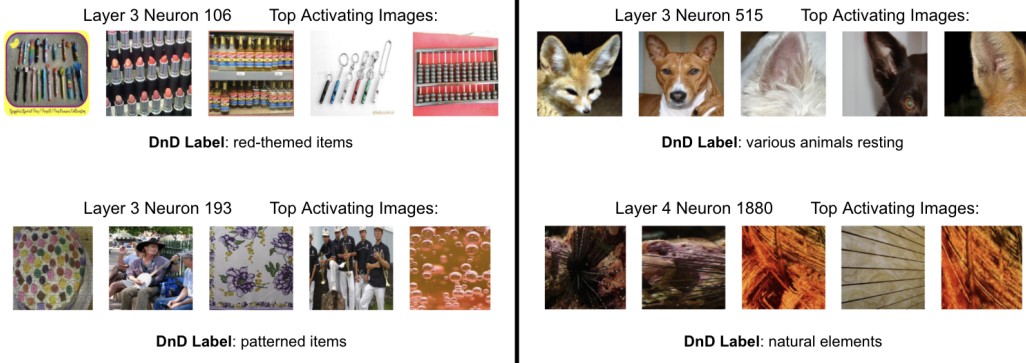

Figure 27: **Failure cases of DnD**. We showcase specific neuron examples from dissecting ResNet-50 that exemplify instances where **DnD** can produce ambiguous or inaccurate labels.

### A.10 Limitations

One limitation of Describe-and-Dissect is the relatively high computational cost, taking on average about 38.8 seconds per neuron with a Tesla V100 GPU. However, this problem can be likely well-addressed as the current pipeline has not been optimized for speed-purposes yet (e.g. leveraging multiple GPU etc). Another potential limitation of our method is that since it first converts the images into text and then generates its labels based on the text, **DnD** likely cannot detect purely visual elements occasionally found in lower layers of networks, such as contrast or spatial elements. Additionally, our method only takes into account the top $k$ most highly activating images when generating labels. Excluding the information found in lower activating images may not allow our method to gain a full picture of a neuron. However, this problem of only focusing on top activations is shared by all existing methods and we are not aware of good solutions to it. Finally, **DnD** is limited by how well the image-to-text, natural language processing, and text-to-image models are able to perform. On the other hand, this also means that future innovations in these types of Machine Learning models can increase the performance of our method.

**Polysemantic neurons:** Existing works Olah et al. (2020); Mu & Andreas (2020); Scherlis et al. (2023) have shown that many neurons in common neural networks are polysemantic, i.e. represent several unrelated concepts or no clear concept at all. This is a challenge when attempting to provide simple text descriptions to individual neurons, and is a limitation of our approach, but can be somewhat addressed via methods such as adjusting **DnD** to provide multiple descriptions per neuron as we have done in the Appendix A.7. We also note that due to the generative nature of **DnD**, even its single labels can often encapsulate multiple concepts by using coordinating conjunctions and lengthier descriptions. However, polysemantic neurons still remain a problem to us and other existing methods such as Bau et al. (2017), Hernandez et al. (2022), and Oikarinen & Weng (2023). One promising recent direction to alleviate polysemanticity is via sparse autoencoders as explored by Bricken et al. (2023).

**Out-of-Distribution Synthetic Images:** While generated synthetic images in Step 3 (Best Concept Selection) may be out of distribution, we note that an explanation should explain the behavior of the model on all inputs, not just in-distribution examples. A neuron explained by "car" should activate on all kinds of cars, not just in-distribution ones, otherwise, a better explanation would be more specific.

**Challenges in comparing different methods:** Providing a fair comparison between descriptions generated by different methods is a very challenging task. The main method we (and previous work) utilized was displaying highly activating images and asking humans whether the description matches these images. This evaluation itself has some flaws, such as only focusing on the top-k activations and ignoring other neuron behavior. In addition, the question of what are the highly activating images and how to display them is a surprisingly complex one and requires multiple judgment calls, with different papers using different methods with little to no justification for their choices. First, which images are most highly activating. When our neuron is a CNN channel, its activation is 2D – sorting requires summarizing to a scalar, typically using either max or average pooling. In our experiments we used average, but max gives different sets of results. Second, the choice of probing dataset is important, as it will affect which images are displayed. Finally choosing if/how to highlight the highly activating region to raters will have an effect. We chose not to highlight as we crop the highly activating regions which can provide similar information. If highlighting is used, choices like what activation threshold to use for highlights and how to display non-activating regions will also affect results. Different choices for all these parameters will generate varying results. Consequently, it is hard to provide fully fair comparisons, so results such as Section 4.3 should be taken with a grain of salt.

### A.11 Broader Impact

Our work aims to improve our understanding of machine learning models, which we expect to have positive societal impact as more understanding will decrease the chance of harm from deploying unsafe or unreliable models. However, one potential downside may be that if explanations look convincing but are not reliable enough, they may cause a user to place unwarranted trust on an unreliable model. Our methods can be used to help reveal and mitigate biases in existing vision models by, for example, identifying neurons detecting sensitive attributes such as skin-color.

