# OpenReview forum: "Interpreting Neurons in Deep Vision Networks with Language Models"
_TMLR — Accepted by TMLR_

### Review · Reviewer_Vcjh · 2024-11-07

**Summary Of Contributions:**

The paper presents a novel XAI technique called Describe-and-Dissect (DnD). The technique describes the "role" of hidden neurons in ResNets by generating "concepts" that best describe images that lead to high activations of these hidden neurons.

The network is first probed with random crops of images from a probing set. The crops leading to largest activations of neurons are then captioned using BLIP ("candidate concept generation"). These candidate concepts are then selected by generating images using Stable Diffusion and scoring the concepts by the resulting activation of these hidden neurons ("best concept selection").

The authors compare the effectiveness of the proposed method with other methods (Network Dissection, CLIP-Dissect, MILAN), and then also examine the effect of neuron pruning based on the outputs of the method with two land-cover prediction networks.

**Audience:**

Yes

**Broader Impact Concerns:**

The authors claim that "One important role of interpretability tools is the ability to create real-world impacts" in Section 5.

This aspect might be elaborated in more detail in a "Broader Impact Statement". I see this though more of an opportunity for the authors if they want to highlight this aspect of the work, and not as a requirement.

**Claims And Evidence:**

Yes

**Requested Changes:**

Critical requests:

1. The PDF is missing the Appendix, but the Appendix is referenced in many places.
1. Section 3.4 "Scoring Function" should be revised for improved clarity (the other parts of Section 3 are easy to follow). Some examples of what I found unclear in this section: Are `t` and `β` free parameters? How are they set? Is the "average rank" the defined as $\text{avg}(\mathcal{H}_j)$? What does it mean that "Rank($R_j$) sorts $R_j$"?

Improvements to Section 5:

1. Making list of NAIP and EuroSAT labels more prominent would help readability of this section
1. Make it clear which model is evaluated in which subsections.
1. Section 5.2. claims that the results after pruning are "better than the baseline" – but the difference is very small, and there is no information on statistical variance.
1. In Table 7, the observed changes should also be compared to pruning a random subset of neurons. Otherwise the observation that "human interpretable neurons account for more critical roles" does not hold.
1. The first paragraph of Section 5.1 is not clear: how is the "spatial mean" computed from two text embeddings? how are the neurons grouped together by using a single threshold? what was the prompt used with GPT (which version?) for "one-shot classification into the 6 NAIP superclasses"?
1. Section 5.1. claims that "clusters containing more neurons are frequently associated with more interpretable neurons" – looking at the concepts in Figure 4, I don't find this finding obvious
1. in Section 5.3. it is not clear how "Term Frequency Analysis" is used, or how the "relationship" is studied
1. Not sure what Section 5.3 shows. It seem that using probing images from ImageNet and Broden has lead to concept generation that is not useful for the EuroSAT dataset, and that this leads to a rather random selection of two neuron groups, one of which has a large effect, and the other does not.

Requests to strengthen the submission:

1. Show some failure cases and examine why they happen and how they could be improved.
1. Add a code link so the community can use the method.
1. Add FALCON results to Figure 1, and compare with FALCON in Section 4.2
1. Missing table of intermediate neurons (Section 4.1.1) – Table 5 only has results of FC layer
1. Add results for a non-ResNet model, or explain why this is impossible.
1. Ablate BLIP and Stable Diffusion models – how sensitive is the method to these models?
1. what `K` was used? (Section 3.2)
1. Clarify (Section 4.3) "we used the images calculated by our method" – does this mean images from $\mathcal{D}_\text{probe}$? are these the same 10 images that were presented to workers to rate the description? (asking because the same paragraph mentions "10 images" twice, but only mentions the "images calculated by our method" on the second occurrence)
1. Add the prompt that was used for summarization (Step 2B)


Typos

1. The abbreviation "WPMI" is used before it's spelled out as "weighted pointwise mutual information".
1. Section 3.1 "and an perform".

**Strengths And Weaknesses:**

Strengths:

1. The presented method is simple and training-free, using widely available off-the-shelf models.
1. The method performs better than previous methods, as shown in the quantitative preference evaluation by raters.
1. Overall, the paper is well written and easy to follow.

Weaknesses:

1. The paper only discusses the application of the method to ResNets.
1. The method is missing some ablations, such as the used image-to-text model or the text-to-image model, as well as the used thresholds. Having good ablations of these parameters would give confidence that the method is tuned sufficiently to be used by the community.
1. I found the Use Case Section hard to follow and I'm not convinced by the results presented in that section. This section makes the submission harder to read and adds little insight in its current form.

---

> ### Author Response · Authors · 2024-12-15
> **Author response to Reviewer Vcjh (1/2)**
>
> Dear Reviewer Vcjh,
>
> Thank you for your feedback!
>
> __#1 Appendix__
>
> We would like to clarify that we originally included the Appendix in the supplementary material, but have included it in the revised main submission pdf to make it easier to follow.
>
> The Appendix includes comprehensive ablation studies of different stages in our pipeline shown in Section A.4.
>
> __#2 Application beyond ResNets__
>
> We note that DnD can be applied to *any* deep vision network, not just ResNets. We provide additional qualitative experiments on a ViT-B-16 model in section A.2.
>
> __#3 Clarification on Section 3.4__
>
> Regarding section 3.4, $t$ and $\beta$ are free parameters set to $t$ = 10 and $\beta$ = 5 for the purposes of our experiment. The average rank is defined as $R_j$, and “Rank($R_j$) sorts $R_j$” means that Rank(⋅) sorts the average rank $R_j$ of each candidate concept in increasing order. The scoring functions are discussed in more detail in Appendix A.1.4. We have edited the manuscript to clarify the notation.
>
> __#4 Clarification on Section 5__
>
> Thank you for the feedback. We have made the following clarifications regarding section 5.
>
> __a. Evaluated Models__
>
> Sections 5.1 and 5.2 are evaluating a Tile2Vec ResNet-18 while section 5.3 is evaluating a EuroSAT-trained ResNet-50 model. We have edited the manuscript to clarify which model is being evaluated.
>
> __b. “Better than baseline” after pruning:__
>
> To address your concern regarding our claim in section 5.2 that the results after pruning are "better than the baseline,” we want to clarify that the purpose of section 5.2 is to identify large portions of neurons that do not contribute to the final classification and show this by pruning those neurons and displaying how the average accuracy barely changes. As such, the difference in the results before and after pruning being very small supports our point and the purpose of this section. However, thank you for pointing out our wording error of “better.” We have updated section 5.2 to more accurately say “better than or equal to performance to the baseline.”
>
> __c. Pruning Random Neurons__
>
> Thank you for the suggestion! We have added random pruning baseline results to Table 7 and supplemental results in Appendix A.5. We outline the edits below:
>
> 1. Retrained Classifier Results
>
> The Tile2Vec model evaluated contains two parts: 1. A ResNet model to compute embeddings of land tiles, 2. a random forest classifier to predict the land coverage of the output embedding. Using the originally proposed Tile2Vec pipeline, we first prune neurons in the ResNet model and then train the classifier on embeddings of the pruned model. Because we retrain the classifier on the pruned pipeline, the difference in accuracy using DnD pruning and that of random pruning is fairly minor, but we can see random pruning consistently suffers an accuracy drop while DnD pruning does not. Results are shown below:
>
> | | Layer 1 | Layer 2 | Layer 3 | Layer 4 | Layer 5|
> |--|---------|-----------|---------|----------|--------|
> | DnD Acc. (%) | 71.07 | 71.54 | 71.75 | 72.04 | 71.76 |
> | Random Pruning Acc. (%) | 69.3 | 68.96 | 70.13 | 69.95 | 70.99 |
>
> 2. Fixed Classifier Results
>
> We also evaluate the Tile2Vec model using a fixed classifier. We train the classifier using embeddings from the full unpruned pipeline and evaluate by fixing the classifier while pruning the ResNet model only. The table shows results using a similar percentage of pruned neurons as Table 7 in the manuscript.
>
> | | Layer 1 | Layer 2 | Layer 3 | Layer 4 | Layer 5|
> |--|---------|-----------|---------|----------|--------|
> | DnD Acc. (%) | 51.0 | 49.5 | 70.5 | 56.0 | 60.5 |
> | Random Pruning Acc. (%) | 45.0 | 47.0 | 66.6 | 49.5 | 56.0 |
> | % Pruned | 23.44 | 50.0 | 26.95 | *50.78 | *56.64 |
>
> *Due to GPT’s generative captioning, the similarity between neuron labels can vary between runs, leading to slightly different numbers of neurons pruned
>
> __d. Additional experiment details__
>
> We calculate the similarity between labels as the dot product between their word embeddings. We have updated the manuscript to clarify this. Regarding the method we use to group neurons, we use a single threshold value to measure the CLIP similarity between two neuron descriptions. We place both neurons in the same group if their similarity is larger than the set threshold. The prompt used for one-shot-classification is outlined in Appendix A.1.3. All experiments are conducted using GPT-3.5 turbo.
>
> __e. Relationship between cluster size and neuron interpretability__
>
> Regarding Figure 4, we note that many concepts with low cluster size have labels that are abstract (ie. “Abstract Nature Patterns”) or too nuanced (ie. “Corporate Beachside Retreat”). In contrast, we see more informative concepts in groups with more neurons (like “Green Striped Patterns”).

---

> > ### Author Response · Authors · 2024-12-15
> > **Author response to Reviewer Vcjh (2/2)**
> >
> > __f. Term Frequency Analysis__
> >
> > In section 5.3, Term Frequency Analysis refers to identifying the frequencies of words or phrases in a body of text. In our case, we identify which neuron DnD label appears the most often. The “relationship” refers to how these identified neurons affect the model’s classification task.
> >
> > __g. Section 5.3 Characterizing Spurious Correlations__
> >
> > To clarify the purpose of section 5.3, we label neurons in the model and show through pruning that neurons representing a certain concept (pink/purple) are important for the classification task while neurons representing another concept (fishing) have no impact on the final cropland cover label prediction. Although the descriptions are not tailored to EuroSAT, our results show that certain spurious correlations are related to the model’s internal decision-making.
> >
> > __#5 Failure Cases__
> >
> > Thank you for the suggestion! We have included qualitative examples of specific DnD failure cases in Appendix A.9. Failure cases stem primarily from vague descriptions (“patterned items”) and spurious image captioning (“red-themed-items”). These drawbacks occur most often on uninterpretable or polysemantic neurons, which is also a limitation for many contemporary works. We address this issue in more detail in Appendix A.10 Limitations.
> >
> > __#6 Code Release__
> >
> > We will make all code publicly available prior to publication.
> >
> > __#7 Add Comparison to FALCON__
> >
> > Following your suggestion to add comparisons to FALCON, we added some neuron examples in appendix section A.2 comparing our method to FALCON labels. Unfortunately, due to lack of time and errors in the FALCON code documentation, we could not yet run another crowdsourced experiment to compare FALCON with our method. As such, we have not yet modified our Figure 1 and Section 4.2.
> >
> > __#8 Missing Table of Intermediate Neurons (Section 4.1.1)__
> >
> > Because only the final layer output neurons have ground truth labels (corresponding to ImageNet 1k classes), we are unable to extend the experiment to intermediate layer neurons. We assess these intermediate neurons using human evaluations, presented in Section 4.3 Crowdsourced Experiment.
> >
> > __#9 Results on non-ResNet models__
> >
> > We included results for ViT-B-16 in Appendix A.2 Figure 13 of our original manuscript and updated the section with additional evaluations on a larger ResNet-152 model in Appendix A.2 Figure 14.
> >
> > __#10 Ablation studies -  BLIP and Stable Diffusion__
> >
> > We present an ablation study for BLIP in Section 4.4.1 DnD with a Fixed Concept Set and present additional failure cases in Appendix A.4.2. Instead of generative captioning, we use CLIP to select the best neuron label from a set of predefined words–we used the 20k most common English words. Additionally, we also experiment with BLIP2 in Appendix A.4.3 and observe similar results as BLIP with notable exceptions, presented in Figure 20.
> >
> > We also ablate Stable Diffusion by removing the Best Concept Selection (step 3) of our pipeline in Section 4.4.2 and show quantitative results in Table 6. The full pipeline (DnD w/ Concept Selection) equals or outperforms the ablated version (DnD w/o Concept Selection) across all 4 layers of ResNet-50.
> >
> > __#11 Choice of K parameter__
> >
> > The K used in section 3.2 was 10. Using a higher K provides inference over a larger sample of activating images, but increases computational cost in image captioning and GPT summarization.
> >
> > __#12 Clarification on Probing Dataset (Section 4.3)__
> >
> > In section 4.3, "we used the images calculated by our method" refers to the 10 images presented to the workers. These images are not Dprobe, they are the top 10 highly activating images from the imageset DprobeDcropped.
> >
> > __#13 Summarization Prompt (Step 2B)__
> >
> > We have included the prompt used for summarization in Appendix A.1.3 of the original manuscript.

---

> > > ### Comment · Reviewer_Vcjh · 2024-12-20
> > >
> > > Appendix: It was an oversight on my side about the "missing" appendix. Thank you for adding the appendix to the PDF, it makes for a very interesting reading, and also gives clear answers to a number of the points I raised in my original review. Since the appendix is rather voluminous, feel free to submit it as a separate document again (though I think it is nice to have it in the main PDF to increase its visibility).
> > >
> > > Scoring functions: The appendix also helps with understanding the scoring function better (though I did not notice where exactly you have "edited the manuscript to clarify the notation."). I could find β=5 in the appendix, but I couldn't find where you specify t=10.
> > >
> > > Section 5.2: Thank you for adding the random baseline and expanding the last part of the paragraph. But I can still read "while achieving a better result than the baseline (no neurons pruned).", even though you say that you amended the text to "better than or equal to performance to the baseline." in your reply.
> > >
> > > → Can you double check that the manuscript is indeed at the **latest revision**?
> > >
> > > I noticed that you added "Through extensive qualitative, quantitative, and use-case analysis, we show that DnD
> > > outperforms prior work by providing higher-quality neuron descriptions, greater generality and flexibility, and
> > > significant potential for social impact." to the **conclusion**. How exactly do you back up the claim for "significant potential social impact"?
> > >
> > > As for the ablations, I have read Section 4.4.1 and the ablation of step 3. What I meant is to compare different models to see how sensitive the method is to a particular model, i.e. comparing BLIP with another generative text model, and comparing Stable Diffusion with another generative image model. Section A.4.3 indeed adds some interesting findings in this regard, even though the two examples don't show conclusively that BLIP is indeed better than BLIP-2.
> > >
> > > Thank you already for implementing most of the requested changes (failure cases, random baseline etc), I think they make the submission significantly stronger. I understand that the request for adding FALCON to Figure 1 and Section 4.2 was a stretch in view of the additional rater work required and will not insist on this change.
> > >
> > > (I only expect answers to the points outlined in **bold**, but feel free to reply to the other comments as well.)

---

> > > > ### Author Response · Authors · 2024-12-21
> > > > **Author response to Reviewer Vcjh**
> > > >
> > > > Dear Reviewer Vcjh,
> > > >
> > > > We appreciate your additional feedback!
> > > >
> > > > **#1 Latest revision**
> > > >
> > > > Thank you for pointing this out! We have fixed the statement to “better than or equal to the baseline performance” and specified $t = 10$ in Section 3.4 Scoring Function of our latest draft. We also highlighted additional clarifications to the scoring function notation in blue.
> > > >
> > > > **#2 Conclusion**
> > > >
> > > > Land cover prediction is a useful application to study sustainability and climate change, which can have a significant social impact. For this task, DnD can identify spurious correlations that improve performance which helps to ensure reliability and user trust for ML models. Our method can also be extended to other socially impactful tasks such as medical imaging and facial recognition tasks.
> > > >
> > > > We hope this addresses your concerns and would like to thank you again for your feedback!

---

### Review · Reviewer_zajy · 2024-11-08

**Summary Of Contributions:**

The paper presents Describe-and-Dissect (DnD), a method for generating natural language descriptions of hidden neurons in vision networks. The DnD pipeline has 3 main steps:
- Probing Set Augmentation: Building a set of images to record neuron activations on before generating a description for the neuron
- Candidate Concept Generation: Given a neuron, feed the highly-activating images to BLIP to generate descriptions and use GPT-3.5 Turbo for summarizing neuron descriptions
- Concept Selection: Given possible neuron descriptions (concepts), synthetic images are generated for each concept and fed back to the model to select the most representative concept.

The method is evaluated through qualitative and quantitative studies, ablation analyses, and a use-case application in land cover prediction.
The reviewer encourages the authors to revise the writing carefully and provide missing key details.

**Audience:**

Yes

**Claims And Evidence:**

No

**Requested Changes:**

- Clarify holistic vs. local concepts: Provide a clear distinction between holistic and local concepts to contextualize the importance of including holistic concepts in neuron interpretation. Explain how DnD captures both types and why holistic concepts matter for understanding neuron function.
- Justify D_probe and D_cropped: In Step 1, clearly describe how D_probe is constructed and its purpose, along with its limitations, and justify the inclusion of D_cropped. Without these explanations, Step 1 lacks sufficient grounding for technical evaluation.
- Potential OOD activation in step 3: Discuss the potential for OOD activation when using synthetic images in Step 3. Consider exploring alternative approaches or at least acknowledging this as a limitation in the Discussion section to provide a balanced view of DnD’s applicability.
- Explain assumptions on last layer neurons in Section 4.2.1: Provide justification for assuming that last-layer neurons correspond to class labels. Clarifying this assumption is necessary for readers to interpret the validity of the textual similarity evaluation results.
- Qualitative analysis: I like the AMT human study setup, yet, I hope authors can give more statistics about the numbers (e.g. std for mean rating) in Tables. Also, showing qualitative examples where each method is the best should be insightful to know where DnD works/does not work.

- Threshold justification in section 5.1: In Section 5.1, provide rationale for the threshold value (0.8) used to determine concept similarity/dissimilarity in land cover prediction.

- Revise writing for consistency and clarity: Correct typos and ensure stylistic consistency across the paper, such as standardizing “Figure” and “Table” usage. Additionally, combining tables with limited rows (e.g., Tables 3 and 4) would streamline presentation. Examples of typos in writing: (a) compare descriptions similarity to the class name that neuron is detecting; (b) For a pair of candidate concepts sets. Also, the Limitations are placed in Appendix A.8 should be moved to the main text (the authors could briefly discus the main points to save spaces).

- Provide additional experiment details:

(a) In Section 4.2.1, include details on the experimental setup, such as the number of neurons evaluated and input specifics for the textual similarity function.

(b) Clarify the concept selection approach in Section 4.4.2 for cases where Best Concept Selection is not used.

(c) Add statistical details for Tables, such as standard deviation for mean ratings in the AMT study.

**Strengths And Weaknesses:**

Strengths
- Innovative approach to pruning: The pruning experiments in Section 5 are a compelling application of DnD
- Interesting human evaluation: The AMT human study is well-designed, and the results demonstrate DnD’s interpretability benefits

Weaknesses
- Lack of clarity on holistic vs. local concepts: The paper mentions the importance of holistic versus local concepts, especially when discussing Kalibhat et al. (2023). However, this distinction is not clearly defined, leaving it unclear about what makes holistic concepts valuable or how they are incorporated.
- D_probe and D_cropped: Details on how D_probe is constructed and why D_cropped is needed are lacking. This absence makes it difficult to assess the technical correctness of Step 1 and its effect on downstream steps. The paper would benefit from explicitly stating the role of D_probe and its limitations, followed by a justification of D_cropped. I looked into A.1 in Appendix and did not find that info.
- Potential Out-of-Distribution (OOD) activation in step 3: In Step 3, feeding synthetic (generated) images into the model can return OOD activations because the model could see the generated images drawn from a different distribution than the distribution of its training set. This could make the activation not reliable to evaluate the generated concepts. The reviewer encourages the authors to explore this limitation (i.e. Do we have better alternative than generating synthetic images and feed back to the model? I know that there are a bunch of papers following the same approach but it has the OOD limitation as mentioned. I would love to see the authors discuss this problem and possible solution in Discussion/Limitations.)
- Last-layer neurons in evaluation: In Section 4.2.1, the authors assume that last-layer neurons correspond to class names, yet it is unclear why this would be the case. In [1,2], people show evidence that the deep layers of NNs are responsible for high-level concepts (e.g. nose, tail), that together form the object, but not the class name. Without a clear justification, this assumption may be problematic, as high-level concepts do not always equate to specific class labels.
- Writing and consistency issues: Typos, inconsistencies (e.g., “Figure” vs. “Fig.,” “Section” vs. “Sec.”), and a lack of stylistic coherence impact readability. Tables with only a few rows, such as Tables 3 and 4, could be combined to improve presentation.
- Lack of experiment details and justification in key sections:

** In Section 4.2.1, the setup for the textual similarity evaluation is missing key details, such as the number of neurons evaluated and input specifics for the similarity function.

** Section 4.4.1 does not clarify how concepts are generated without using an image-to-text model, which would be crucial information for replication.

** In Table 5, the minimal performance gain from including the image-to-text model raises questions about its value. The 2.6% when using mpnet is not enough evidence to demonstrate the importance of the img-to-text model.

** In Section 4.4.2, it is unclear how concept selection is achieved if not using Best Concept Selection. The authors do not specify how they chose a concept among the generated ones.

[1] Visualizing and Understanding Convolutional Networks

[2] Understanding Neural Networks Through Deep Visualization

---

> ### Author Response · Authors · 2024-12-15
> **Author response to Reviewer zajy (1/2)**
>
> Dear Reviewer zajy,
>
> Thank you for the feedback! Below we respond to your specific concerns.
>
> __#1 Clarification on holistic vs local concepts__
>
> We would like to clarify that holistic concepts are emergent concepts that are represented by the whole image rather than just a part of it. Local concepts are concepts that are contained to a portion of the image and can be identified with just this portion. In regards to our discussion about Kalibhat et al. (2023), we noted that since their method observes only the cropped portions of an image, they can only identify local concepts and can miss important holistic concepts that represent the neuron.
>
> In particular, this is helpful when a neuron activates on a more nuanced concept or activates on a highly focused region in probing images. We show two examples in Appendix A.1.2 Figure 6. Under both conditions, including non-cropped images helps the image-to-text models better identify target concepts and avoid vague or spurious descriptions.
>
> __#2 Clarification on D_probe vs D_cropped__
>
> We would like to clarify that $D_{probe}$ is the probing image set, or the set of images that we feed into the target deep vision network to measure its neuron activations with. For many of our experiments, we use the union of the ImageNet validation dataset and Broden as $D_{probe}$. $D_{cropped}$ is then the set of image crops resulting from cropping the images in $D_{probe}$ based on which portions of the images caused the neurons to activate the most (attention cropping). The details of our attention cropping can now be found in section A.1.1. The justification for $D_{cropped}$ can be found in its ablation study in section A.4.1.
>
> __#3 Concern regarding generated images being Out of Distribution__
>
> Thank you for the suggestion, this is a valid concern and we have added discussion on this in limitations. However, we do not believe this is a major issue, especially if we are using a good image generation model. This is because ideally we hope that an explanation would explain the behavior of the model on all inputs, not just in-distribution examples. A neuron explained by “car” should activate on all kinds of cars, not just in-distribution ones, otherwise, a better explanation would be more specific.
>
> __#4 Clarification regarding last layer neurons__
>
> We would like to clarify that the last layer neurons we refer to are in the output layer of the model, not hidden layers. Each neuron in the output layer is trained to activate only when the specific class is present in the input, and can therefore be described with the corresponding class name.
>
> __#5 Qualitative analysis__
> Following your suggestion, we have added the standard deviation to the AMT study tables and provided further qualitative examples of DnD in Appendix A.2. Additionally, we have added examples of DnD failure cases in Appendix A.9.
>
> __#6 Threshold justification__
> We determined the threshold value of 0.8 through experimentation. A high threshold value corresponds to more ungrouped neurons (and larger percentage of neurons pruned), while low thresholds correspond to less ungrouped neurons. We find our model achieves the best accuracy vs. percentage pruned performance at around 0.8. The hyperparameter can be changed depending on the model evaluated.
>
> __#7 Writing and consistency__
> Thank you for the suggestions, we have followed your feedback to improve our writing and table style.

---

> > ### Author Response · Authors · 2024-12-15
> > **Author response to Reviewer zajy (2/2)**
> >
> > __#8 Experimental Details__
> >
> > Thank you for the suggestion, below we describe the requested details, which we have also clarified in the manuscript:
> >
> > __Section 4.2.1__
> >
> > We compute the CLIP and mpnet cosine similarity between embeddings of generated neuron descriptions and ground truth labels. Higher similarities represent more accurate neuron descriptions. We evaluate output layer neurons because each neuron corresponds to the predicted ImageNet class (clarification #4 above). The experiment is conducted across all 1000 FC Layer output neurons. The BERTScore metric measures the semantic interpretability of descriptions.
> >
> > __Section 4.4.1__
> >
> > To evaluate the impact of generative image captioning, we replace BLIP with OpenAI’s CLIP model while keeping the remaining pipeline fixed. We provided the CLIP model with each neuron’s top activating images along with a set of potential labels to select from. For this experiment, potential labels are the set of 20,000 most common English words. We calculate the dot product between each word’s text embedding ($L(t_i)$) and the top activating images’ embeddings ($E(I_m)$). The best neuron label is the word with the maximum dot product. Because CLIP uses predefined concepts, there is often a lack of expressiveness in neuron labels when compared to generative BLIP captioning, which we presented in Appendix A.4.2.
> >
> > __Table 5__
> >
> > Thank you for the observation! We’d like to clarify that Table 5 presents results after ablating BLIP and replacing the image-to-text model with CLIP. Although the decrease in performance is minor, CLIP requires users to provide a set of predefined concepts. Image-to-text models are fully generative, meaning they don’t require concept sets. This is significant because predefined concepts restrict the expressibility of interpretability models to only words within the set. In particular, restricting explanations to a predefined set makes it impossible to discover concepts we did not expect to see, while it is possible with a generative method like ours.
> >
> > Additionally, the changes in terms of embedding cosine similarity (i.e. mpnet) can sometimes be misleading as the scores do not cover the entire [-1, 1] range and instead return values in a smaller interval i.e. [0.2, 0.5] regardless of the text pair used, which makes differences look smaller than they are.
> >
> > __Section 4.4.2__
> >
> > We select the first concept produced by GPT in Concept Generation (Step 2) and keep the remaining pipeline constant.

---

### Review · Reviewer_63zx · 2024-12-01

**Summary Of Contributions:**

This paper proposes a training-free framework, named Describe-and-Dissect (DnD), to interpret and explain neurons in deep networks. The proposed DnD leverages large multimodal models to provide natural language descriptions for neurons. As the proposed method is training- free, it can be easily adapted to general models. Besides, the proposed method overcomes the existing limitation by providing more complex and higher-quality descriptions rather than representing concepts with simple descriptions. Authors have conducted extensive experiments and ablation studies and results can validated the effectiveness of the proposed method.

**Audience:**

Yes

**Broader Impact Concerns:**

Authors have not included any broader impact discussions, I encourage authors include necessary discussions.
1. When leveraging large language models, are there any biased contained in these models?
2. Since the proposed method is providing explanations for neurons, authors should discuss use cases that includes detecting features related to humans (e.g. facial recognition).

**Claims And Evidence:**

Yes

**Requested Changes:**

I have concerns which are expected to be resolved by the authors listed in the following:

1. At the beginning of section 3, authors can briefly explain the motivations (1 short sentence) when introducing each step.
2. In step 2, this paper utilizes both the BLIP and GPT models to generate candidate concepts, which is less flexible because it requires 2 large models (Table 5 shows that the improvements with BLIP is not sigfinicant). I wonder have authors tried to replace GPT model with similar language models in BLIP family or even re-use text tower in BLIP?
3. I wonder authors have further scaled the method with larger ResNet models as well as variants, e.g., VGGNet, etc?
4. In Table 7, I am curious the efficiency given by introducing the pruning operations? Since authors show that the certain percent of neurons have been prunes, whether it can bring some speedup?
5. One open discussion question: is the proposed method limited to CNNs? Have authors consider any extensions in transformers?

**Strengths And Weaknesses:**

Overall, the paper clearly explain the proposed method, experiments and discussions.
1. Related work section is clear and comparisons with existing method is presented clearly.
2. Authors break down the proposed method into several subsections and introduce each step clearly.
3. Experimental results can validate that using the proposed method can give the superior performance compared with existing method.
4. Authors have included extensive experimental results in supplementary materials.

---

> ### Author Response · Authors · 2024-12-15
> **Author response to Reviewer 63zx**
>
> Dear Reviewer 63zx,
>
> Thank you for your feedback! Below we respond to your requested changes:
>
> __#1. Motivations for each step__
>
> Following your suggestions, we have included a brief summary of the motivation behind each step in our pipeline in Sec 3 in the updated manuscript (changes highlighted in blue color). Specifically:
> - For __Step 1. Probing Set Augmentation__, we highlighted that this step is to better describe localized neuron activations.
> - For __Step 2. Candidate Concept Generation__, we highlighted that this is the main step generating a set of possible explanations.
> - For __Step 3. Best Concept Selection__, we highlighted that this step refines the prediction by selecting the most accurate neuron description from all descriptions produced in Step 2.
>
> __#2. Replacing BLIP+GPT with multimodal GPT__
>
> Following your suggestion of combining Step 2’s two large models into one, we conducted an additional experiment by exchanging BLIP + GPT in Step 2 of DnD with the image caption capabilities of GPT-4o. Results show that due to the large overlap of the GPT models, performance is similar in many examples. However, the original DnD pipeline seems to have the edge on DnD w/ GPT-4o when BLIP’s image captioning capabilities out-perform GPT-4o’s. Though including 2 large models in Step 2 may reduce the flexibility of our pipeline, it can increase performance by specifying fine-tuned models for each designated task. We have included this in Appendix Section A.4.4.
>
> __#3 Results describing neurons in Larger models__
>
> In response to your suggestion, we have conducted additional experiments on ResNet-152 in Appendix A.2 Figure 14. Our model performs similarly even when scaled to larger vision networks.
>
> __#4 Model speedup after pruning__
>
> Regarding our Use Case experiments, we have also shown an increase in model efficiency after pruning. We observed a moderate (13.63%) increase in speed after performing pruning on the first layer of Tile2Vec ResNet-18 and further gains are possible if running inference more optimized for sparse models.
>
> __#5. Our proposed pipeline for different architectures (e.g. transformers)__
>
> Thank you for the question! Our pipeline is compatible with Vision Transformers. We have provided the results of ViT-B-16 in the original manuscript and supplementary materials, as described in Section 4.1 *“We qualitatively analyze results of randomly selected neurons from various layers of ResNet-50, ResNet-18, and ViT-B-16. Sample results are displayed in Figure 1 and Figures 8, 9, 10, 11, 12, 13, and 14 in the Appendix”*. The results are shown in Appendix A.2 Figure 13.
>
> In general, our method is not limited to the architecture and works for modern architecture such as CNNs and Vision Transformers.
>
>
> __#6 Broader Impact__
>
> Thank you for the feedback, we have included the following Impact Statement in Appendix A.11.
>
> Our work aims to improve our understanding of machine learning models, which we expect to have positive societal impact as more understanding will decrease the chance of
> harm from deploying unsafe or unreliable models. However, one potential downside may be that if explanations look convincing but are not reliable enough, they may cause a
> user to place unwarranted trust on an unreliable model. Our methods can be used to help reveal and mitigate biases in existing vision models by for example identifying neurons detecting sensitive attributes such as skin-color.

---

> > ### Comment · Action_Editor_RP3o · 2024-12-27
> > **Discussion and recommendation.**
> >
> > Dear Reviewer 63zx,
> >
> > The authors appear to have answered your questions in detail. Please make sure to read the rebuttal before submitting your official recommendation. Feel free to ask any last minute questions as well.
> >
> >
> > Best wishes,
> >
> > AE

---

> > ### Comment · Reviewer_63zx · 2024-12-31
> >
> > Thanks authors for providing thorough response.
> >
> > 1. Please double check the order of the figures in the Appendix, e.g., "additional experiments on ResNet-152" is in A.2 or A.3.
> >
> > 2. As you mentioned that "Our model performs similarly even when scaled to larger vision networks.", does it mean there is a scaling bottleneck because we expect to more improved performance with larger models?

---

> > > ### Author Response · Authors · 2025-01-31
> > > **Author response to Reviewer 63zx**
> > >
> > > Dear Reviewer 63zx,
> > >
> > > Thank you for the comment!
> > >
> > > 1. We present additional DnD qualitative examples on ResNet-152 in A.2 Figure 14. In section A.3 we provide a comparison to MILANNOTATIONS also evaluated on ResNet-152.
> > > 2. Our comment that DnD "performs similarly even when scaled to larger vision networks" is meant to convey that experiments show our method is agnostic to target model size. We achieve similar high performance on larger models like ResNet-152.

---

### Decision · Action_Editor_RP3o · 2025-01-02

**Recommendation:** Accept as is

**Comment:**

This paper introduces a new method, Describe and Dissect (DnD) to assign textually describable "concepts" to individual neurons in DNNs. The idea is to identify the most highly activated images (along with augmented, cropped images obtained through self attention), then use Image-to-Text model such as BLIP, together with GPT, to summaries the similarities between the generated images in one caption. The result is then fed through a stable diffusion to generate new synthetic images corresponding to the each candidate concept. Crucially, the best performing candidate concept is then extracted by verifying that it activates the target neuron strongly.

I find the success of this pipeline convincing, especially due to the final candidate concept selection step, which guarantees that the LLM-generated concepts are not off topic/hallucinatory. It is also clear that substantial effort has gone into designing the appropriate scoring functions for that last step. Lastly, the experiments are very extensive, all reviewers recommend acceptance, and the paper is reasonably well written.


Minor points:

As mentioned by reviewer zajy, there is only one subsection in section 4.2, resulting in awkward formatting, which can be improved during the camera-ready stage.

I would also recommend adding an additional pointer to Appendix A.1.5 in Section 3.2 (fourth step).

**Audience:**

The topic is well within TMLR's key areas.

**Claims And Evidence:**

After the reviewing period, the revised paper includes not only convincing illustrations of the visual of the method, but also very comprehensive results on class-wise accuracy, failure cases and ablation studies.